# Guanidine hydrochloride reactivates an ancient septin hetero-oligomer assembly pathway in budding yeast

Courtney R Johnson[1], Marc G Steingesser[1], Andrew D Weems[1], Anum Khan[2], Amy Gladfelter[2], Aurélie Bertin[3,4], Michael A McMurray[1]*

[1]Department of Cell and Developmental Biology, University of Colorado Anschutz Medical Campus, Aurora, United States; [2]Department of Biology, University of North Carolina at Chapel Hill, Chapel Hill, United States; [3]Laboratoire Physico Chimie Curie, Institut Curie, PSL Research University, CNRS UMR 168, Paris, France; [4]Sorbonne Université UPMC Univ Paris 06, Paris, France

**Abstract** Septin proteins evolved from ancestral GTPases and co-assemble into hetero-oligomers and cytoskeletal filaments. In *Saccharomyces cerevisiae*, five septins comprise two species of hetero-octamers, Cdc11/Shs1–Cdc12–Cdc3–Cdc10–Cdc10–Cdc3–Cdc12–Cdc11/Shs1. Slow GTPase activity by Cdc12 directs the choice of incorporation of Cdc11 vs Shs1, but many septins, including Cdc3, lack GTPase activity. We serendipitously discovered that guanidine hydrochloride rescues septin function in *cdc10* mutants by promoting assembly of non-native Cdc11/Shs1–Cdc12–Cdc3–Cdc3–Cdc12–Cdc11/Shs1 hexamers. We provide evidence that in *S. cerevisiae* Cdc3 guanidinium occupies the site of a 'missing' Arg side chain found in other fungal species where (i) the Cdc3 subunit is an active GTPase and (ii) Cdc10-less hexamers natively co-exist with octamers. We propose that guanidinium reactivates a latent septin assembly pathway that was suppressed during fungal evolution in order to restrict assembly to octamers. Since homodimerization by a GTPase-active human septin also creates hexamers that exclude Cdc10-like central subunits, our new mechanistic insights likely apply throughout phylogeny.

*For correspondence: michael.mcmurray@cuanschutz.edu

Competing interests: The authors declare that no competing interests exist.

## Introduction

Septin proteins are found in nearly every eukaryotic lineage, with the exception of land plants (*Auxier et al., 2019*; *Nishihama et al., 2011*; *Onishi and Pringle, 2016*; *Pan et al., 2007*). In most extant organisms studied to date, multiple septin proteins co-assemble into linear, rod-shaped hetero-oligomers with at least some potential to polymerize into filaments (*Fung et al., 2014*; *Mostowy and Cossart, 2012*; *Oh and Bi, 2011*). Septin hetero-oligomers are now known to contribute functionally to a wide variety of cellular processes (*Dolat et al., 2014*; *Saarikangas and Barral, 2011*). A number of human diseases and disorders, ranging from cancer to male infertility, have been linked to septin dysregulation or mutation (*Dolat et al., 2014*; *Saarikangas and Barral, 2011*), but within our understanding of septin roles in cellular function one common theme emerges: the ability to assemble into quaternary complexes is key.

All distinct septins share clear homology, but phylogenetic analysis reveals septin groups and subgroups (*Pan et al., 2007*) that can accurately predict which septin occupies which subunit position within hetero-oligomers (*Nakahira et al., 2010*). However, the subtleties of the distinctions between septins that drive assembly of hetero-oligomers with precise subunit organization remain incompletely understood. Moreover, the oligomerization interfaces have not diverged drastically from the ancestral form, as nearly all septins retain the ability to homo-oligomerize in vitro (*Farkasovsky et al., 2005*; *Versele et al., 2004*) whether or not they do so in vivo. How cells

**eLife digest** For a cell to work and perform its role, it relies on molecules called proteins that are made up of chains of amino acids. Individual proteins can join together like pieces in a puzzle to form larger, more complex structures. How the protein subunits fit together depends on their individual shapes and sizes.

Many cells contain proteins called septins, which can assemble into larger protein complexes that are involved in range of cellular processes. The number of subunits within these complexes differs between organisms and sometimes even between cell types in the same organism. For example, yeast typically have eight subunits within a septin protein complex and struggle to survive when the number of septin subunits is reduced to six. Whereas other organisms, including humans, can make septin protein complexes containing six or eight subunits. However, it is poorly understood how septin proteins are able to organize themselves into these different sized complexes.

Now, Johnson et al. show that a chemical called guanidinium helps yeast make complexes containing six septin subunits. Guanidinium has many similarities to the amino acid arginine. Comparing septins from different species revealed that one of the septin proteins in yeast lacks a key arginine component. This led Johnson et al. to propose that when guanidinium binds to septin at the site where arginine should be, this steers the septin protein towards the shape required to make a six-subunit complex.

These findings reveal a new detail of how some species evolved complexes consisting of different numbers of subunits. This work demonstrates a key difference between complexes made up of six septin proteins and complexes which are made up of eight, which may be relevant in how different human cells adapt their septin complexes for different purposes. It may also become possible to use guanidinium to treat genetic diseases that result from the loss of arginine in certain proteins.

assemble specific septin hetero-oligomers to meet functional demands represents a major focus of septin research. More generally, septins provide a powerful model to study the mechanisms by which multisubunit complexes are assembled with high efficiency and fidelity in living cells.

Septins clearly evolved from an ancestral GTPase (*Leipe et al., 2002*) but multiple septins subsequently lost the ability to hydrolyze GTP. Unlike the building blocks of polymers of the other cytoskeletal NTPases actin and tubulin (and their prokaryotic counterparts), in native septin hetero-oligomers most, if not all, of the septin nucleotide-binding pockets are buried within an oligomerization interface, called the G interface (*Sirajuddin et al., 2007*). Particularly for septin complexes in budding yeast, where hetero-octamers polymerize into filaments via the other, so-called 'NC' septin-septin interface (*Bertin et al., 2008*; *Sirajuddin et al., 2007*), burial of GTP in G interfaces within highly stable complexes severely limits the potential for cycles of GTP binding, hydrolysis, and exchange to modulate the assembly and disassembly of septin filaments. Nonetheless, most septins retain sequences clearly corresponding to regions/motifs in other small GTPases that contact bound nucleotide and/or change conformation upon GTP hydrolysis. These include the G1/Walker/P-loop, residues of which contact each of the three phosphates; the G2/Switch I and G3/Switch II loops, which are in proximity to the γ phosphate when it is present but are in distinct locations when it is absent; and the G4 motif, which contacts the base and dictates specificity for guanosine nucleotides. The question of what roles GTP binding and hydrolysis play in septin biogenesis and function has been a major focus in septin research for over two decades (*Mitchison and Field, 2002*).

The baker's yeast, *Saccharomyces cerevisiae*, has been at the forefront of septin research, for a number of reasons. The repertoire of septin proteins expressed in yeast (seven) is much simpler than that of humans (13). Two yeast septins are expressed and function specifically during the process of sporulation (*De Virgilio et al., 1996*; *Fares et al., 1996*; *Garcia et al., 2016*; *Heasley and McMurray, 2016*; *Pablo-Hernando et al., 2008*), a version of gametogenesis, leaving five septins expressed in nearly stoichiometric amounts (*Kulak et al., 2014*) in mitotically dividing cells: Cdc3, Cdc10, Cdc11, Cdc12, and Shs1. Yeast septin localization during the mitotic cell division cycle is primarily restricted to the cortex, where septins take the form of filamentous rings at the mother-bud neck that undergo a series of discrete structural transitions as the bud emerges and grows and cytokinesis and cell separation take place (*Oh and Bi, 2011*). Septin filament assembly is essential for

proliferation (*McMurray et al., 2011*), hence defects in septin folding and oligomerization translate into clear defects in yeast colony/culture growth. A combination of molecular genetic (*Garcia et al., 2011*; *Iwase et al., 2007*; *McMurray et al., 2011*; *Nagaraj et al., 2008*; *Versele et al., 2004*), biochemical (*Bertin et al., 2008*; *Farkasovsky et al., 2005*; *Garcia et al., 2011*; *Versele et al., 2004*), and structural (*Bertin et al., 2008*; *Farkasovsky et al., 2005*; *Garcia et al., 2011*; *Sirajuddin et al., 2007*) approaches have demonstrated that two kinds of hetero-octamers co-exist, with a linear hexameric core of the order Cdc12–Cdc3–Cdc10–Cdc10–Cdc3–Cdc12 'capped' at each end with either Cdc11 or Shs1. While the functional basis for yeast octamer diversity remains largely unknown, at the level of quaternary structure septins are best understood in *S. cerevisiae*.

We are most interested in the mechanisms of assembly of septin hetero-oligomers, and previously used mutants isolated in unbiased genetic screens to elucidate roles in this process for nucleotide binding (*Weems et al., 2014*) and interactions with cytosolic chaperones (*Johnson et al., 2015*), including the fungal disaggregase Hsp104, a hexameric AAA+ ATPase (*Sweeny and Shorter, 2016*). Here we report that the function of specific septin mutants is affected in unexpected ways by guanidine hydrochloride (GdnHCl), a small molecule used for decades to inhibit Hsp104 ATPase activity in vivo (*Derkatch et al., 1997*; *Jung and Masison, 2001*). Our findings reveal important roles for specific residues and structural motifs in septin hetero-oligomer assembly, suggest a series of key events during fungal septin evolution that enforced incorporation of Cdc10 subunits in hetero-oligomers, and provide what is, to our knowledge, the first evidence that guanidinium (Gdm) has the potential to functionally replace Arg residues in vivo.

## Results

### GdnHCl restores high-temperature septin function to *cdc10* mutants in an Hsp104-independent manner

We previously reported interactions between an ATPase-dead mutant of Hsp104 and a temperature-sensitive (TS) mutant of Cdc10, in which overexpression of the mutant Hsp104 inhibited the proliferation of *cdc10(D182N)* cells at an otherwise permissive low temperature (27°C) (*Johnson et al., 2015*). We expected that the addition of 3 mM GdnHCl to the culture medium would mimic these inhibitory effects in cells with wild-type (WT) Hsp104. Instead, we found complete rescue by GdnHCl of the *cdc10(D182N)* TS proliferation defect at all temperatures tested (up to 37°C) (*Figure 1A*). GdnHCl did not have the same effect on cells carrying TS alleles of other septins, although in its presence a *cdc3(G365R)* strain grew noticeably worse at 27°C, 30°C and 34°C, and a *cdc12(G247E)* strain grew slightly worse at 30°C and slightly better at 37°C (*Figure 1A*). To our surprise, GdnHCl rescue of *cdc10(D182N)* did not require Hsp104 (*Figure 1B*). Others have suggested that a mitochondrial Hsp104 homolog, Hsp78, might share functions with Hsp104 (*Erives and Fassler, 2015*), and Hsp78 and Hsp104 share the same residues within the ATP binding pocket that contact Gdm in the bacterial Hsp104 homolog ClpB (*Figure 1C*), but Hsp78 was also dispensable for GdnHCl rescue of *cdc10(D182N)* high-temperature growth (*Figure 1B*). Inhibition of Hsp104 (i.e., curing of prions) or Hsp78 (i.e., induction of cytoplasmic petites) by GdnHCl in vivo requires concentrations in the medium of ≥1 mM (*Tuite et al., 1981*), but GdnHCl was sufficient to provide partial rescue of 37°C growth by *cdc10(D182N)* cells at concentrations < 0.1 mM (*Figure 1D,E*). We conclude that GdnHCl suppresses septin defects in *cdc10(D182N)* mutants via a mechanism that does not involve Hsp104 or its homolog Hsp78.

### GdnHCl drives exclusion of mutant Cdc10 molecules from higher-order septin structures

We previously noted that the *cdc12(G247E)* mutation appears to decrease the levels of WT Cdc10 (*Johnson et al., 2015*). Cells tolerate loss of Cdc10 by assembling filaments via hexameric building blocks in which Cdc3 forms a non-native G homodimer, but *cdc10Δ* cells are TS, presumably because at high temperature Cdc3 fails to adopt the homodimerization-competent conformation and/or the non-native Cdc3 homodimer is unstable (*McMurray et al., 2011*). Rescue of the TS defects at 37°C of the *cdc10(D182N)* and *cdc12(G247E)* mutants by GdnHCl could reflect improved assembly of Cdc10-less hexamers, rather than any specific effect on the *cdc10(D182N)* allele. Consistent with this idea, GdnHCl also fully suppressed the TS phenotype resulting from a G100E substitution in Cdc10

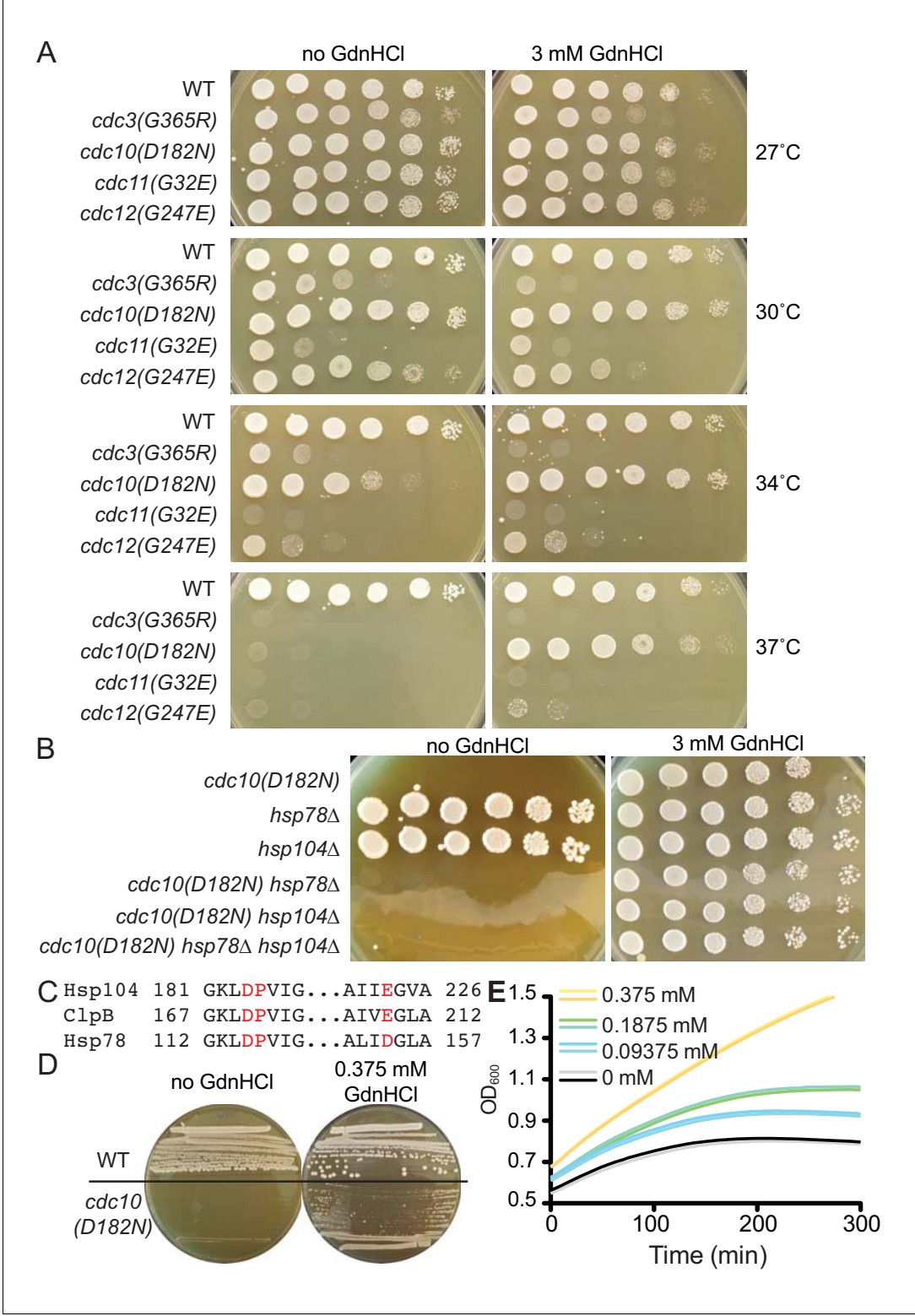

**Figure 1.** GdnHCl rescues the temperature-sensitive defects of *cdc10(D182N)* mutants in a manner that does not require Hsp104 or Hsp78. (**A**) Dilution series of yeast cells spotted on rich (YPD) agar medium with or without 3 mM GdnHCl and incubated at the indicated temperatures for 3 days before imaging. Strains were: BY4741, CBY07236, CBY06417, CBY08756, and CBY05110. Effects of GdnHCl on septin ring formation and on the growth strains carrying other mutant alleles are provided in *Figure 1—figure supplement 1*. (**B**) As in (**A**), but only at 37°C

*Figure 1 continued on next page*

*Figure 1 continued*

and with strains of the indicated genotypes. (C) Amino acid alignment of regions *S. cerevisiae* Hsp104 and Hsp78 with *Thermus thermophilus* ClpB, showing conservation of residues (in red) that contact Gdm in the ClpB crystal structure (PDB 4HSE). (D) As in (A), but at 37˚C and with BY4741 ('WT') or CBY06417 ('*cdc10(D182N)*') cells and the indicated concentrations of GdnHCl. The cells were streaked with a toothpick, rather than spotted from dilutions. (E) Growth curves for the *cdc10(D182N)* strain CBY06417 at 37˚C in the indicated concentrations of GdnHCl. The optical density at 600 nm ('OD600') was measured at 5 min intervals. Each line shows the mean for 12 replicate cultures in the same row of the 96-well plate. Duplicate rows were monitored for each GdnHCl concentration. The online version of this article includes the following source data and figure supplement(s) for figure 1:

**Source data 1.** Optical density readings of culture growth for *Figure 1E*.
**Figure supplement 1.** GdnHCl rescues high-temperature septin function and neck filament assembly in *cdc10* mutants.

(*Figure 1—figure supplement 1A*), as well as that of *cdc10Δ* (*Figure 2A*). Clear but partial GdnHCl rescue was also observed for *cdc10(G44D)* cells (*Figure 1—figure supplement 1A*). We confirmed via immunofluorescence with anti-Cdc11 antibodies (*Figure 1—figure supplement 1B*) and electron microscopy (*Figure 1—figure supplement 1C*) that at 37˚C GdnHCl-treated *cdc10* cells assembled filamentous rings composed of other septins. Thus GdnHCl does not simply bypass the need for septin filaments.

Given these results, we wondered if in cells carrying point-mutant *cdc10* alleles GdnHCl bypasses altogether the incorporation of the mutant Cdc10 subunits during septin hetero-oligomer assembly. Indeed, even at temperatures permissive in the absence of GdnHCl for septin ring incorporation of Cdc10(D182N)-GFP, in the presence of 3 mM GdnHCl the fluorescence of Cdc10(D182N)-GFP was restricted to the cytoplasm (*Figure 2B*). WT Cdc10-GFP continued to localize to the bud neck despite the presence of GdnHCl (*Figure 2B*). Similarly, in diploid cells co-expressing Cdc10-mCherry and Cdc10(D182N)-GFP, the addition of GdnHCl specifically eliminated Cdc10(D182N) incorporation in septin rings, with no obvious effect on WT Cdc10 (*Figure 2C*). Our results suggest that when mutations within the Cdc10 G interface perturb the ability of Cdc10 to acquire a conformation that binds tightly to Cdc3, GdnHCl allows other Cdc3 molecules to outcompete the mutant Cdc10 proteins for occupancy of the Cdc3 G interface, ultimately co-assembling with Cdc11, Cdc12, and Shs1 into Cdc10-less hexamers capable of robust septin functions. As predicted from this model, GdnHCl was unable to rescue *cdc10Δ* growth when Cdc3 carried a mutation (W364A) previously shown (*McMurray et al., 2011*) to block Cdc3 homodimerization (*Figure 2D*).

## GdnHCl promotes Cdc10-less filament assembly in vitro by purified, recombinant complexes

Yeast septin function is regulated (in poorly defined ways) by a large number of pathways (*Longtine and Bi, 2003*), which might be altered by GdnHCl to indirectly promote Cdc3 homodimerization. GdnHCl might also alter other properties of Cdc10-less yeast cells to promote septin function despite inefficient Cdc3 homodimerization. For example, deletion of *DPL1*, which encodes a dihydrosphingosine phosphate lyase, improves the proliferation of *cdc10Δ* cells at the moderate temperature of 30˚C (*Michel et al., 2017*). The authors of that study speculated that the absence of Dpl1 alters the properties of the plasma membrane in ways that mimic the effects of low temperature and allow Cdc10-less septin complexes to better function in cytokinesis (*Michel et al., 2017*). Accordingly, Cdc10-less septin function at high-temperature in the presence of GdnHCl might reflect GdnHCl inhibition of Dpl1. However, *cdc10Δ dpl1Δ* cells were only able to proliferate at 37˚C when GdnHCl was present in the growth medium (*Figure 1—figure supplement 1D*). While these data do not exclude the possibility that GdnHCl alters membrane properties in ways independent of Dpl1, they demonstrate that the GdnHCl effect on septin function in *cdc10Δ* cells is Dpl1-independent.

To ask if GdnHCl is able to promote Cdc10-less septin filament assembly outside of yeast cells and away from membranes, we co-expressed Cdc3, Cdc11, and hexahistidine-tagged Cdc12 (6xHis-Cdc12) in *E. coli*, which lacks native septins, and purified septin complexes as described previously (*McMurray et al., 2011*). We then used negative staining and electron microscopy (EM) to ask if these purified complexes appeared as hexamers in high-salt solution and polymerized into filaments

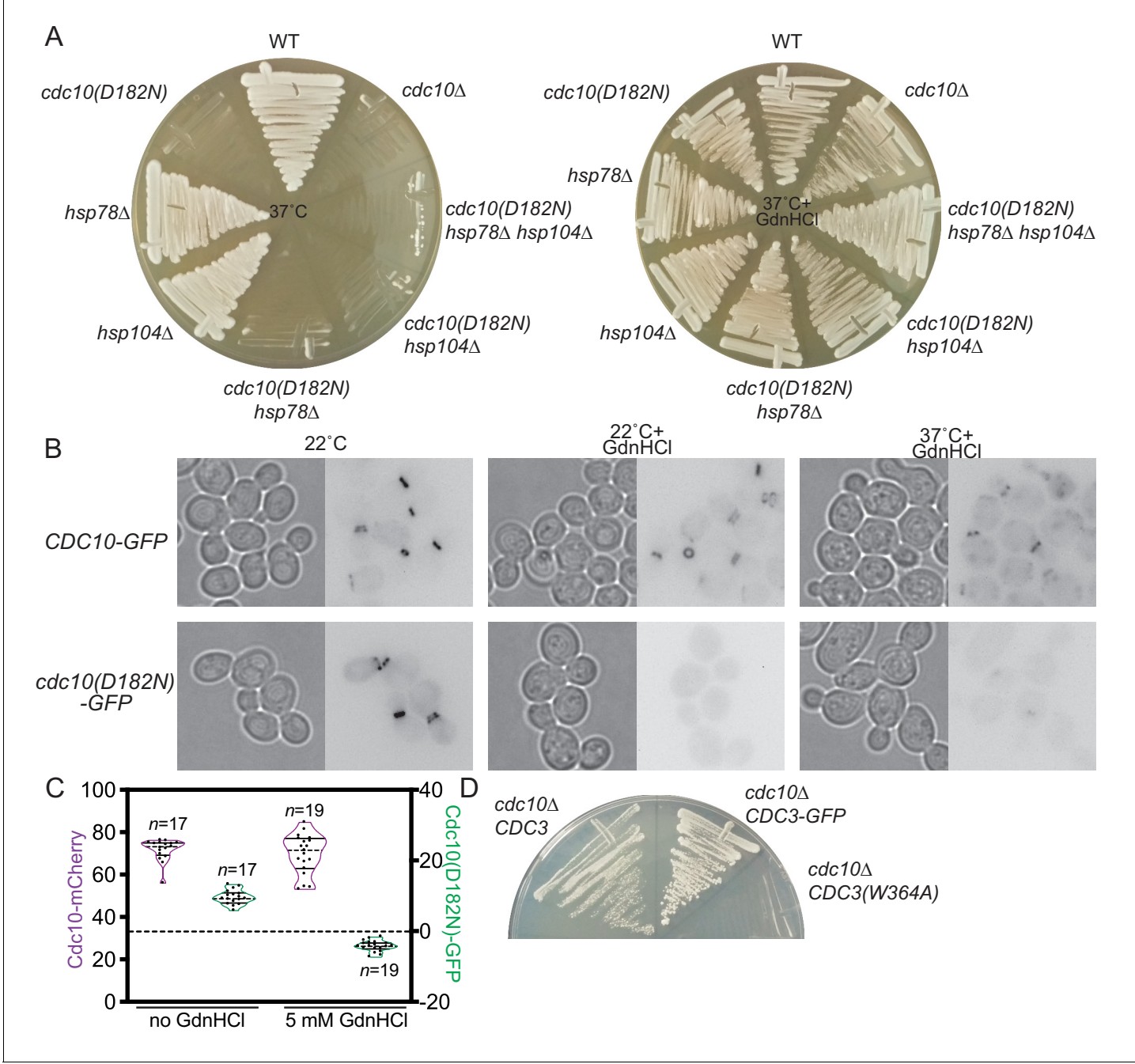

**Figure 2.** GdnHCl promotes exclusion of mutant Cdc10 molecules from higher-order septin assemblies. (**A**) As in *Figure 1B*, but with the addition of *cdc10Δ* cells, and the cells were streaked with a toothpick, rather than spotted from dilutions. (**B**) Cultures of the strains JTY3985 ('*CDC10-GFP*') or JTY3986 ('*cdc10(D182N)-GFP*') were grown overnight in liquid culture at the indicated temperatures with or without 3 mM GdnHCl and then imaged by microscopy with transmitted light (left images) or for GFP fluorescence (right images). Fluorescence images were inverted to improve visibility. (**C**) Cells of the strain JTY4020 were cultured at 37°C with or without 5 mM GdnHCl and imaged as in (**B**), then line scans of bud neck fluorescence were performed for the indicated numbers of cells to quantify levels of Cdc10-mCherry (purple) or Cdc10(D182N)-GFP (green). For each data point, cytosolic signal was subtracted from bud neck signal; negative values thus indicate cytosolic signal greater than septin ring signal. In violin plots, solid lines indicate quartiles and dashed lines are medians. (**D**) In strain JTY5104, the chromosomal sources of Cdc3 and Cdc10 are eliminated by deletion and Cdc3(W364A) and Cdc10 are supplied by *LYS2-* or *URA3*-marked plasmids, respectively. Following introduction of *LEU2*-marked plasmid encoding WT Cdc3 (pFM831) or Cdc3-GFP (pML109), the transformants were passaged on medium with α-aminoadipate to select for loss of the *CDC3(W364A)* plasmid. These clones were then streaked, along with cells of the original strain JTY5104 carrying both plasmids ('*cdc10Δ CDC3(W364A)*'), on medium with 3 mM GdnHCl and 5-fluoro-orotic acid (FOA, to select for loss of the *CDC10* plasmid) and incubated for 4 days at 22°C.

*Figure 2 continued on next page*

*Figure 2 continued*

The online version of this article includes the following source data for figure 2:

**Source data 1.** Intensity values of tagged septins at septin rings for *Figure 2C*.

upon salt dilution. We previously used the same approach to demonstrate that a mutation in the Cdc3 G interface, G261V, promotes septin hexamer and filament assembly in the absence of Cdc10 (*McMurray et al., 2011*). To test effects of GdnHCl, we either grew the bacteria in the presence of 50 mM GdnHCl and kept GdnHCl in all buffers thereafter, or we only introduced the GdnHCl during purification. In the absence of GdnHCl, Cdc10-less complexes were detected during purification via high-salt size-exclusion chromatography (SEC) as a single peak (in addition to a peak of aggregates eluting in the void volume) which, when analyzed by EM, was composed of particles containing two or three septins (*Figure 3A–C*). These complexes were unable to polymerize into filaments in low salt (*Figure 3D*), consistent with our earlier work (*McMurray et al., 2011*). Two-septin particles likely represent Cdc3–Cdc12 hetero-dimers, since in these conditions Cdc11 occasionally dissociates from the ends of hetero-octamers containing Cdc10 (*Bertin et al., 2008*).

By contrast, in SEC purification of complexes synthesized in the presence of GdnHCl a new, higher-molecular weight peak appeared (*Figure 3A*), which EM analysis revealed to contain trimers, tetramers, pentamers and hexamers (*Figure 3B,C*). The larger complexes were frequently 'kinked' at a location three subunits from one end (*Figure 3B*), consistent with weak septin-septin contacts at this point, which we also previously observed with the Cdc3(G261V) mutation (*McMurray et al., 2011*). Upon salt dilution, the Cdc10-less complexes synthesized in the presence of GdnHCl robustly polymerized into bundles of laterally-associated filaments (*Figure 3D*), very similar to the behavior of WT hetero-octamers that contain Cdc10 (*Bertin et al., 2008*). Fourier transform analysis of the striations in these filament bundles revealed a repeating unit of ~27 nm, the length of a septin hexamer (*Figure 3D*). Adding GdnHCl to complexes synthesized in its absence did not allow filament formation (*Figure 3D*). We conclude from these data that GdnHCl is able to promote septin filament assembly by recombinant Cdc10-less septin complexes produced in a heterologous system that lacks septin-regulatory pathways. Thus GdnHCl likely acts directly on septin proteins, and only when it is present during septin folding/assembly.

## Unbiased in silico modeling and phylogenetic analysis point to Thr302 as the site of Gdm binding in Cdc3

We identified the G261V mutation in Cdc3 using an unbiased genetic selection for suppressors of the TS phenotype of *cdc10Δ* cells, and concluded that the introduction of Val in this position stabilizes a conformation of the G interface that self-associates better than the major conformation populated by WT Cdc3 at high temperatures (*McMurray et al., 2011*). We reasoned that Gdm might bind in the Cdc3 G interface and similarly promote the homodimerization-competent conformation. For clues as to where Gdm might bind, we first used the structure of human SEPT2 bound to the non-hydrolyzable nucleotide analog 5'-guanylyl imidodiphosphate (GppNHp) (*Sirajuddin et al., 2009*) to generate a homology model of the globular domain of monomeric Cdc3, and then performed unbiased in silico docking simulations to ask for the lowest-energy predicted sites of Gdm binding. Four sites had binding free energies of $\leq -3$ kcal/mol. One site, between the side chains of Thr302 and His262, lies in the G interface, and had the second-lowest predicted free energy of binding (*Figure 4A*).

As a parallel approach, we considered that (i) Gdm mimics the distal end of an Arg side chain, and (ii) a number of previous in vitro studies of diverse non-septin proteins – including T4 lysozyme (*Baldwin et al., 1998*), β-galactosidase (*Dugdale et al., 2010*), and carboxypeptidase A (*Phillips et al., 1992*) – demonstrated that Gdm is able to functionally occupy molecular vacancies that are created by the replacement of an Arg residue with a residues having a smaller side chain (e.g. Ala). We thus hypothesized that one or more Arg residues might contribute to G interface contacts for other septins that are, unlike *S. cerevisiae* Cdc3, capable of robust G homodimerization.

To identify candidate Arg residues, we performed a phylogenetic analysis of Cdc3 homologs from different species, which we separated into two categories based on the severity of phenotypes accompanying deletion of the *CDC10* homolog. If non-*Saccharomyces* species possess an Arg that

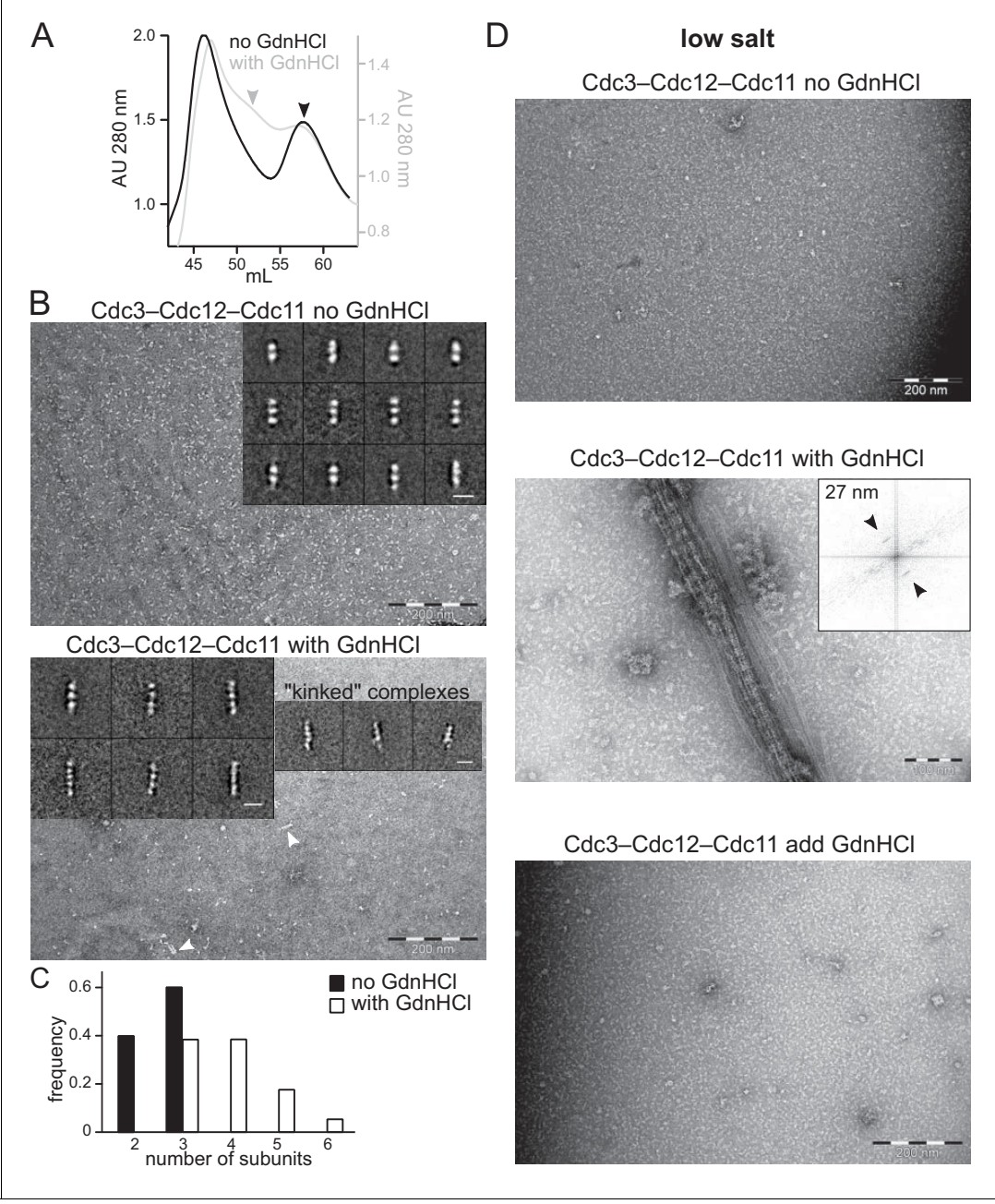

**Figure 3.** GdnHCl promotes higher-order assembly by recombinant Cdc10-less septin hexamers produced in *E. coli*. Complexes purified from *E. coli* cells co-expressing Cdc3, Cdc11, and 6xHis-Cdc12 in the absence or presence of 50 mM GdnHCl, as indicated, were analyzed by size-exclusion chromatography (SEC) in high-salt (300 mM NaCl) buffer, to prevent filament polymerization, and fractions containing the peak of protein, as judged by absorbance at 280 nm, were examined with an electron microscope. (A) SEC elution profiles overlaid to illustrate the appearance of a new peak of larger particles (eluting at ~ 52 mL) in the samples prepared with GdnHCl. Arrowheads indicate the peaks from which fractions were taken for EM analysis. The void volume (~46 mL) contains protein aggregates. 'AU', absorbance units. (B) EM analysis of samples indicated in (A) were exchanged into non-polymerizing high-salt buffer (50 mM Tris-HCl pH 8, 300 mM NaCl with or without 50 mM GdnHCl). Arrowheads point to complexes longer than trimers. Inserts display representative class averages resulting from image processing, with 10 nm scale bars. (C) Frequency distribution of sizes of particles found in high-salt samples, from 4233 particles of the sample produced without GdnHCl and 3247 particles produced with GdnHCl. (D) As in (B) but following dilution into polymerizing ('low salt', 50 mM Tris-HCl pH 8, 50 mM NaCl with or without GdnHCl) conditions. Insert shows the Fourier transform of a bundle of filaments as in the image beneath it, with arrows pointing at diffraction peaks reflecting repetitive distances of 27 nm. The bottom panel shows complexes synthesized and purified in the absence of GdnHCl to which GdnHCl was added to 50 mM prior to dilution in low salt buffer also containing 50 mM GdnHCl.

The online version of this article includes the following source data for figure 3:

**Source data 1.** Absorbance values at 280 nm at each elution volume for *Figure 3A*.

**Source data 2.** Particle size values for *Figure 3C*.

favors G homodimerization by Cdc3, then we predicted this group of species should easily form Cdc10-less septin hexamers and be better able to tolerate deletion of the Cdc10 homolog. Such functional information was available for non-*Saccharomyces* species from eleven distinct fungal genera. In *Aspergillus* (*fumigatis* (**Vargas-Muñiz et al., 2015**) or *nidulans* **Hernández-Rodríguez et al., 2012**), *Candida albicans* (**Warenda and Konopka, 2002**), *Cryptococcus neoformans* (**Kozubowski and Heitman, 2010**), *Fusarium graminearum* (**Chen et al., 2016**), *Neurospora crassa*

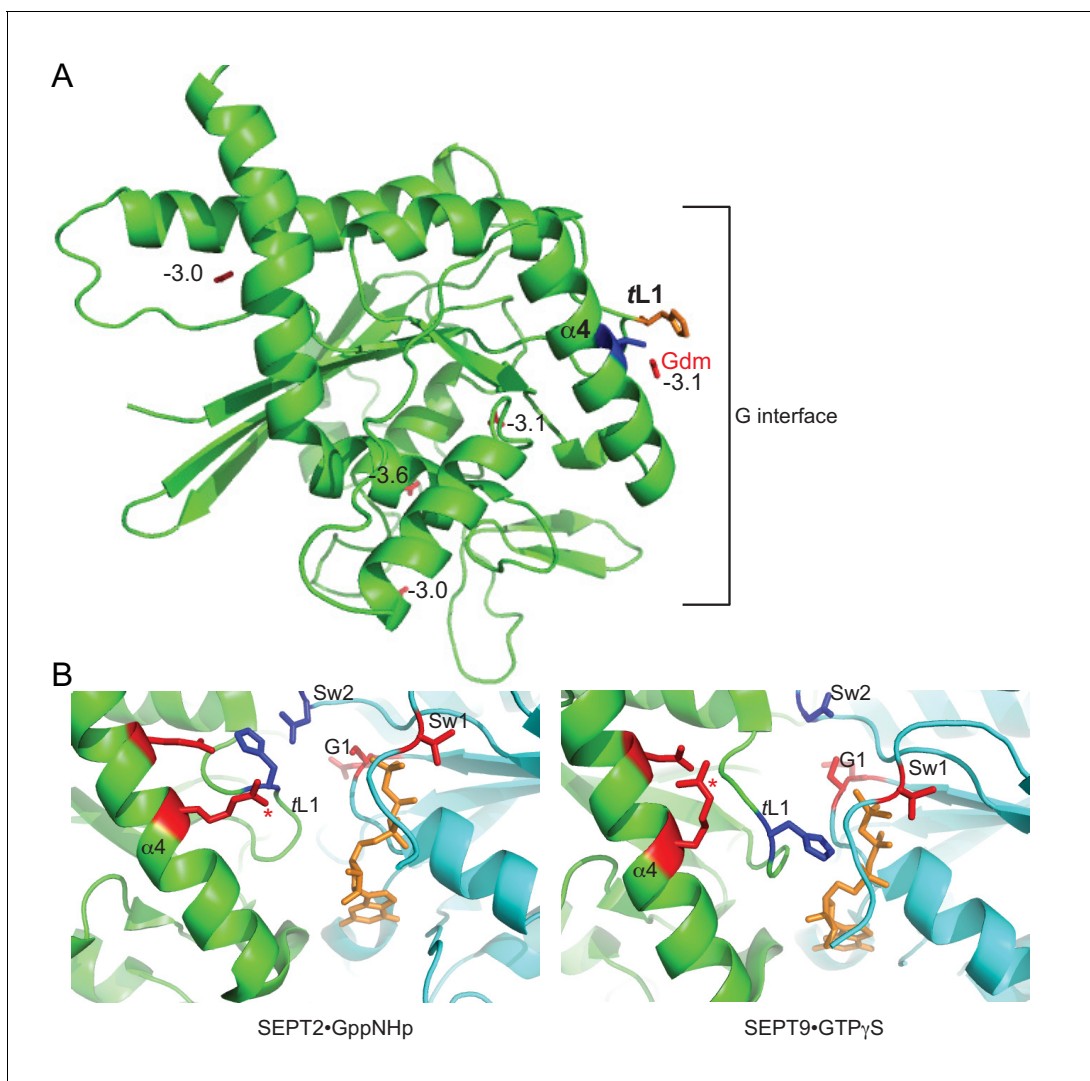

**Figure 4.** Parallel in silico modeling and phylogenetic analysis independently point to Gdm binding near Thr302 and His262 in Cdc3. (**A**) A model of the structure of Cdc3 based on SEPT2•GppNHp (PDB 3FTQ) was used for unbiased in silico docking of Gdm. The top 5 predicted sites of Gdm binding are shown (red), labeled with predicted free energies of binding, in kcal/mol. Thr302 is in blue, His262 in orange. 'α4', α4 helix; '*t*L1', *trans* loop 1. (**B**) Cartoon ribbon view of human septin G homodimer crystal structures showing selected side chains. Red, residues corresponding to the non-conservative changes seen in the phylogenetic analysis in *Figure 4—figure supplement 1B*. Blue, conserved His residue of *trans* loop 1 and Asp residue of Switch II region. Orange, GTP analog. Red asterisk, Arg in α4 helix. 'α4', α4 helix; '*t*L1', *trans* loop 1; 'Sw1', Switch I; 'Sw2', Switch II; 'G1', G1 motif/P-loop.

The online version of this article includes the following figure supplement(s) for figure 4:

**Figure supplement 1.** Co-variation during evolution of Cdc3 residues clustered around the *trans* loop 1 of the G dimerization interface identifies an Arg residue 'missing' in species that poorly tolerate *cdc10Δ*.

(*Berepiki and Read, 2013*), *Schizosaccharomyces pombe* (*An et al., 2004*), and *Ustilago maydis* (*Alvarez-Tabarés and Pérez-Martín, 2010*), deletion of the Cdc10 homolog has distinctly milder phenotypic consequences compared to deletion of the Cdc3 homolog. In *Magnaporthe oryzae*, septin ring assembly is perturbed by deletion of any septin, but higher-order, filamentous structures persist in cells lacking the Cdc10 homolog Sep4, whereas in the other mutants septin localization is almost exclusively diffuse (*Dagdas et al., 2012*). Finally, in *Coprinopsis cinerea*, a UV-induced mutant defective in fruiting body development was rescued by a gene encoding CcCdc3, but the expression of CcCdc10 remained about 100-fold decreased in the mutant following rescue, suggesting that the mutant was defective in both CcCdc3 and CcCdc10 expression, and that rescue of CcCdc3 expression was sufficient to restore function (*Shioya et al., 2013*). We interpret these results as evidence that in *C. cinerea*, as well, loss of Cdc10 is better tolerated than loss of Cdc3.

Non-*Saccharomyces* yeasts in the family Saccharomycetaceae behave differently (see *Figure 4—figure supplement 1A*). Only the Shs1 homolog is non-essential for proliferation in *Kluyveromyces lactis* (*Rippert and Heinisch, 2016*). In *Ashbya gossypii*, deletion of any mitotic septin prevents septin ring assembly; loss of the Cdc10 homolog is as severe as loss of the Cdc3 homolog (*Helfer and Gladfelter, 2006*). According to our hypothesis, if there is a key Arg residue that promotes Cdc3 homodimerization, and *S. cerevisiae* lacks it, then *K. lactis* and *A. gossypii* should also lack it.

*Figure 4—figure supplement 1B* shows an alignment of protein sequences for Cdc3 homologs from the fungal species listed above. As ScCdc3 is predicted to be unable to hydrolyze GTP (*Sirajuddin et al., 2009*), we also included for comparison human SEPT2 and SEPT9, because for these septins crystal structures are available of non-native G homodimers in the 'GTP state' (bound to the non-hydrolyzable GTP analog GppNHp or GTPγS, respectively). Only 16 residues were distinctly different between a group including *S. cerevisiae* and the two species that cannot tolerate Cdc10 loss, and the group of species that tolerate *CDC10* deletion. For six of these residues, the changes reflect substitutions with strong predicted effects on amino acid properties (polarity, charge, size, etc.). One of these non-conservative variants lies in the C-terminal extension, which is disordered in septin crystal structures. Based on the crystal structures of SEPT2•GppNHp and SEPT9•GTPγS, all of the other five non-conservative variants cluster near the G interface (*Figure 4B*), and one of these residues – corresponding to Thr302 in ScCdc3 – is an Arg in eight of the nine species that tolerate Cdc10 loss (*Figure 4—figure supplement 1B*). Moreover, in the SEPT2•GppNHp structure the Gdm group of the corresponding Arg in SEPT2 is located in the same place where our in silico docking results predicted that Gdm binds to Cdc3, between Thr302 and His262 (*Figure 4*). Together, these findings strongly support the idea that Gdm promotes Cdc3 homodimerization by occupying the same site in the Cdc3 G interface that in other septins is occupied by the Gdm moiety of an Arg side chain. This Arg residue was likely substituted to Thr during the evolution of the fungal lineage that includes *S. cerevisiae*, *A. gossypii*, and *K. lactis*, concomitant with loss of Cdc10-less hexamers in these yeasts.

## GdnHCl promotes higher-order septin assembly in *A. gossypii* cells lacking Cdc10

Our phylogenetic comparisons predicted that Gdm should act similarly on the *A. gossypii* Cdc3 homolog to promote AgCdc3 homodimerization in the absence of Cdc10. To test this prediction, we exposed *A. gossypii* cells lacking *CDC10* to GdnHCl and assessed higher-order septin assembly by monitoring the localization of Shs1-GFP expressed in the same cells. As reported previously (*Helfer and Gladfelter, 2006*), in *cdc10Δ* cells in the absence of GdnHCl Shs1-GFP fluorescence was cytoplasmic and diffuse (*Figure 5*). Addition of GdnHCl restored the localization of Shs1-GFP to branch points and hyphal tips (*Figure 5*), the same pattern observed in WT cells (*Helfer and Gladfelter, 2006*). Thus the ability of GdnHCl to promote septin function in the absence of Cdc10, and the loss of Arg at the position corresponding to ScCdc3 Thr302, are both conserved over >100 m years of evolution.

## Chemical rescue of *cdc10* by Gdm derivatives

To better understand the molecular details of Gdm 'rescue' of Arg-substituted mutants, earlier in vitro Gdm studies also tested other small molecules with similar chemical properties, including urea and the GdnHCl derivatives aminoguanidine (Pimagedine, here 'aGdnHCl') and N-ethylguanidine

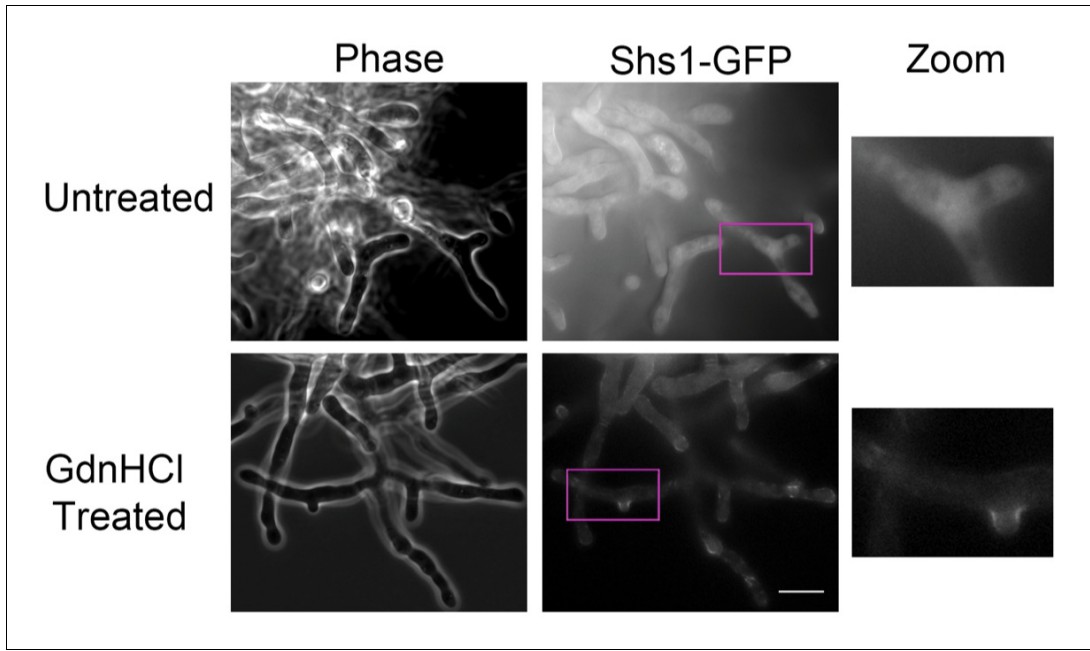

**Figure 5.** GdnHCl rescues *cdc10Δ* in *A. gossypii*. *AgSHS1-GFP Agcdc10Δ* strain AG135 was grown with shaking for 15 hr at 37°C in complete (AFM) medium, after which the culture was split in two and GdnHCl was added to one aliquot to 30 mM final concentration. Cells were imaged after an additional 3 hr growth with shaking. Purple rectangles indicate the approximate region enlarged at right ('Zoom'). Scale bar, 10 μm.

('eGdnHCl') (*Rynkiewicz and Seaton, 1996*). We undertook an analogous approach to study GdnHCl rescue of the *cdc10(D182N)* TS phenotype.

First, we tested urea, which differs from Gdm by only a few atoms (*Figure 6A*). Both GdnHCl and urea denature proteins at high concentrations (>5 M) and, at subdenaturing concentrations (0.05–4 M), both can non-specifically stabilize proteins in vitro, likely by lowering conformational entropy (*Bhuyan, 2002*). If GdnHCl rescues *cdc10* mutants via non-specific conformational stabilization, urea should do the same. We grew WT or *cdc10(D182N)* cells on solid rich medium containing 0.64 M urea, a concentration we found to slow, but not prevent, growth of WT cells, and saw no *cdc10 (D182N)* rescue (*Figure 6A–B*). To test a range of concentrations, we also placed filter disks soaked with GdnHCl or urea on lawns of WT or *cdc10(D182N)* cells. We saw a clear zone of rescue around the GdnHCl disk, but not with urea (*Figure 6C*). These results point to a mechanism of *cdc10* rescue by GdnHCl that is mechanistically distinct from non-specific protein stabilization at subdenaturing conditions.

We next exposed WT or *cdc10(D182N)* cells to 0.375 mM GdnHCl, aGdnHCl, eGdnHCl, or combinations of GdnHCl plus eGdnHCl or GdnHCl plus aGdnHCl (0.375 mM each). At 0.375 mM GdnHCl provided only a partial rescue (*Figure 6D*), allowing us to detect subtle effects of the GdnHCl derivatives when combined with GdnHCl. At 22°C, 0.375 mM of any single drug had no noticeable effect on growth; slight growth inhibition of both WT and mutant cells was observed when the total concentration of GdnHCl plus derivative was 0.75 mM (*Figure 6D*). At 37°C, in addition to the expected partial *cdc10(D182N)* rescue by GdnHCl, there was a very slight rescue by eGdnHCl, and an even less pronounced rescue by aGdnHCl (*Figure 6D*). By contrast, mixing GdnHCl with aGdnHCl or eGdnHCl to a total concentration of 0.75 mM resulted in full or nearly full rescue of *cdc10(D182N)* 37°C growth, respectively (*Figure 6D*). We interpret these findings to mean that aGdm and eGdm (the guanidinium ion derivatives) occupy the position between Cdc3 Thr302 and His262 differently than does Gdm. They may not provide the appropriate molecular contacts across the G dimer interface, or they may bind less well, or even too well (if Gdm acts only transiently during Cdc3 folding). Indeed, aGdm and eGdm were predicted by in silico modeling to bind in the same location as Gdm but with lower free energies, with the additional moieties projecting in various directions (*Figure 6E*) in ways that might further alter Cdc3 conformation. It is also possible

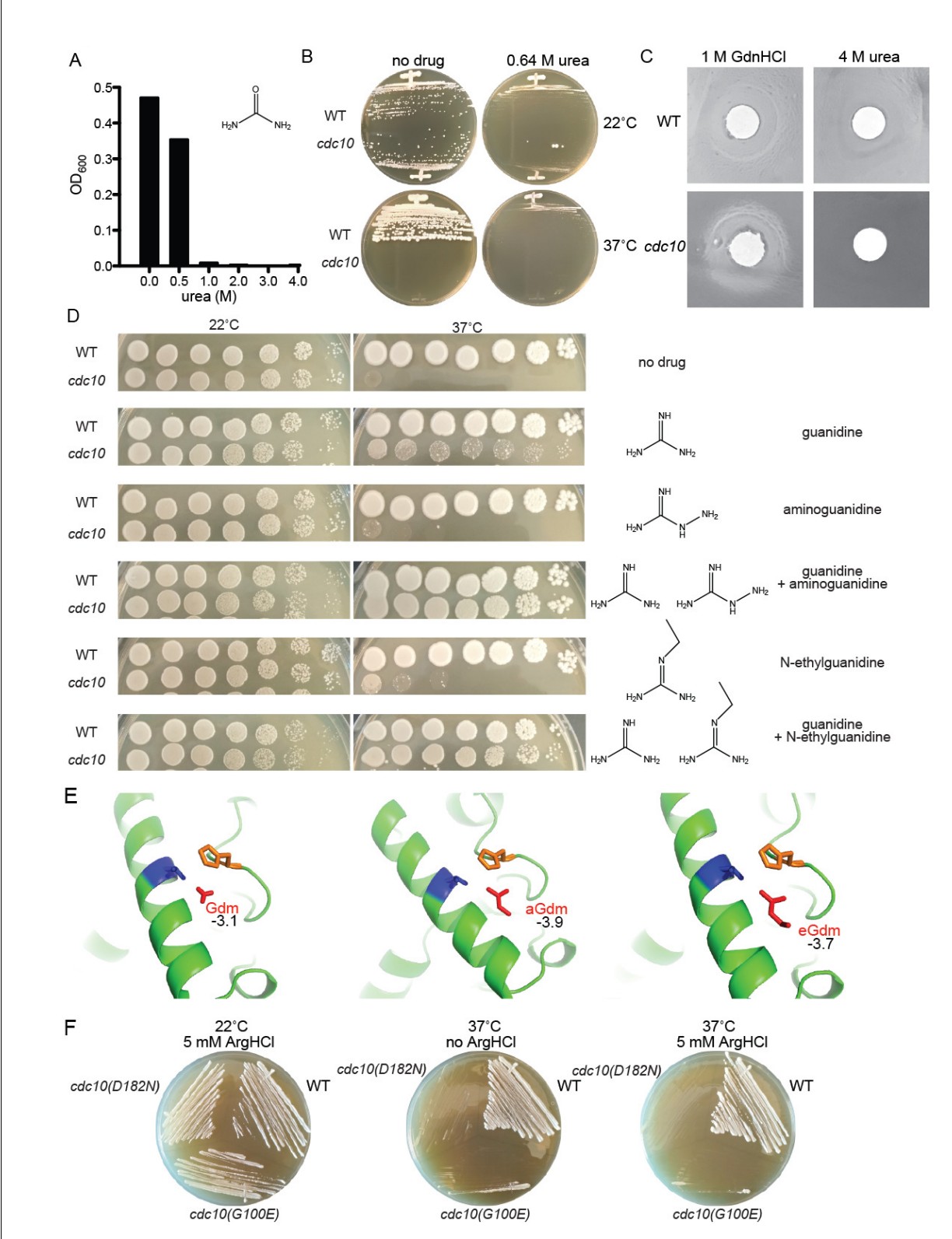

**Figure 6.** Partial rescue of TS defects by GdnHCl derivatives. (**A**) WT strain BY4741 was cultured overnight in rich (YPD) medium containing the indicated concentrations of urea, and the density of each culture was determined with a spectrophotometer. (**B**) BY4741 ('WT') or the *cdc10(D182N)* strain CBY06417 ('*cdc10*') were streaked on plates containing no or 0.64 M urea and incubated at the indicated temperature for 3 days before imaging. (**C**) Cells from saturated YPD cultures of strains used in (**A**) were plated to form a monolayer on the surface of a rich (YPD) agar plate, and a 1-cm-

*Figure 6 continued on next page*

*Figure 6 continued*

diameter filter disk spotted with 5 µL of the indicated drug was placed in the center of the plate. The plate was imaged after incubation at 37°C for 3 days. (D) As in *Figure 1A*, for strains of the indicated genotypes grown on medium containing 0 or 0.375 mM of each indicated drug. (E) As in *Figure 4A*, but zoomed into the area between Thr302 and His262, and including the Gdm derivatives aminoguanidine ('aGdm') and N-ethylguanidine ('eGdm'). (F) As in (B), but with the addition of strain CBY06420 ('*cdc10(G100E)*') and with arginine hydrochloride instead of urea.

that yeast cells are less permeable to these GdnHCl derivatives. Urea did not rescue at all, possibly because it cannot form a cation. Finally, as would be expected if only small molecules like Gdm can fit into the pocket provided by Thr302 and surrounding residues, 5 mM arginine hydrochloride in the medium provided no rescue to *cdc10(D182N)* or *cdc10(G100E)* mutants (*Figure 6F*).

## Mutating Cdc3 Thr302 to Arg prevents *cdc10Δ* rescue by GdnHCl

To test if the absence of Arg at ScCdc3 position 302 is sufficient to explain the inability of *S. cerevisiae cdc10Δ* cells to proliferate at 37°C, we replaced Thr302 with Arg. The double-mutant *cdc10Δ cdc3(T302R)* cells were, like *cdc10Δ CDC3⁺*, TS, but unlike *cdc10Δ CDC3⁺* the addition of GdnHCl failed to restore proliferation at 37°C (*Figure 7*). By contrast, when we replaced Thr302 with Val – an amino acid with properties similar to Thr, including a short side chain – we saw rescue by Gdm (*Figure 7*). We interpret these data as evidence that Gdm binds to ScCdc3 near Thr302 (or Val302) in order to promote homodimerization, but does so in a way that is not recapitulated by an Arg side chain. Instead, Arg302 blocks functional Gdm binding, providing further support for the idea that Gdm acts 'locally', and not via a 'global' structural stabilization mechanism. We conclude that the T302R substitution alone cannot 'reverse evolution', yet GdnHCl does. How?

## Gdm at ScCdc3 position 302 likely 'tunes' a key His residue within the G homodimer interface

ScCdc3 Thr302 is predicted to lie within the septin α4 helix, which is located near both the nucleotide binding pocket and the G interface (*Figure 4*). To begin to understand the molecular mechanism by which Gdm at this position promotes Cdc3 homodimerization, we first examined 13 available septin crystal structures and asked which other residues are nearby the residue in the position equivalent to Thr302. The only non-α4-helix residue within 5 Å in every structure was a highly conserved His (*Table 1*) residing in a loop at the end of the β4 strand previously dubbed the '*trans* loop 1' (*Weirich et al., 2008*), the same His predicted by our in silico modeling to contact bound Gdm in Cdc3 (*Figure 4A*). In SEPT2•GppNHp, His158 contacts a highly conserved Asp within the Switch II loop of the other protomer (*Figure 4B*) (*Sirajuddin et al., 2009*). By contrast, in the sole *Chlamydomonas reinhardtii* septin, CrSEPT, (*Pinto et al., 2017*) or in human SEPT9•GTPγS, the *trans* loop 1 His instead contacts the nucleotide bound by the other protomer (*Figure 4B*). Thus, depending on which septins are involved, this key His residue makes either of two distinct contacts across the septin G dimer interface. If during hetero-oligomerization two alternative septins (e.g. Cdc3 or a mutant Cdc10) present to Cdc3 'competing' G interfaces in distinct conformations, then the position

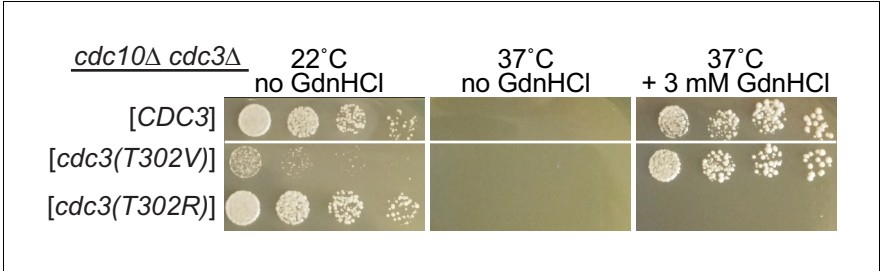

**Figure 7.** Arg at position 302 in Cdc3 blocks rescue of *cdc10* by GdnHCl. As in *Figure 2D*, strain JTY5104 was transformed with a plasmid encoding GFP-tagged Cdc3 (pML109) or derivatives in which the residue at position 302 was mutated from Thr to Val or Arg, followed by selection for loss of the plasmids encoding wild-type Cdc3 and Cdc10. Serial dilutions of the resulting strains were spotted on rich (YPD) medium at the indicated temperatures and containing GdnHCl where indicated, and imaged after 3 days.

**Table 1.** Residues in close proximity to the positions corresponding to ScCdc3 Thr302 or His262 for other septins with solved crystal structures.

| PDB entry, Septin•nucleotide (oligomeric state) | Residues within 5 Å (capitalized) of position corresponding to ScCdc3 Thr302* | Residues within 5 Å (capitalized) of His corresponding to ScCdc3 His262 |
|---|---|---|
| 2QNR, HsSEPT2•GDP (G dimer) | trans loop 1 (pfg**H**Glkp) | α4 (re**R**Lkk**R**ildE) |
| 2QA5, HsSEPT2•GDP (G dimer) | trans loop 1 (pfg**H**glkp) | α4 (rerLkk**R**ilde) |
| 3FTQ, HsSEPT2•GppNHp (G dimer) | trans loop 1 (pfg**H**Glkp) | α4 (rerLkk**R**ildE), Switch I (kie**R**tvq)[†], Switch II (dtpgyg**D**ain)[†] |
| 4Z51, HsSEPT3•GppNHp (monomer) | trans loop 1 (pt**GH**Slrp) | α4 (kseFkq**R**vrkE) |
| 3SOP, HsSEPT3•GDP (G dimer) | trans loop 1 (pt**GH**Slrp) | α4 (kseFkq**R**vrke),, P-loop (gq**S**Glgk)[†], Switch I (vl**P**Ktve)[†] |
| 4Z54, HsSEPT3•GDP (G dimer) | trans loop 1 (pt**GH**Slrp) | α4 (kseFkq**R**vrkE), P-loop (gq**S**Glgk)[†], Switch I (ee**K**lPKtve)[†] |
| 3T5D, HsSEPT7•GDP (G dimer) | trans loop 1 (psg**H**Glkp) | α4 (qqFkk**Q**im), P-loop (ge**S**Glgk)[†] |
| 3TW4, HsSEPT7•GDP (G dimer) | trans loop 1 (psg**H**G**L**kp) | α4 (qqFkk**Q**im), P-loop (ge**S**Glgk)[†] |
| 5CYO, HsSEPT9•GDP (G dimer) | trans loop 1 (at**GH**Slrp) | α4 (rvhFkq**R**ltad), P-loop (gq**S**Glgk)[†], Switch I (ee**R**lPKtie)[†] |
| 5CYP, HsSEPT9•GTPγS (G dimer) | trans loop 1 (atg**HSL**rp) | α4 (rvhFkq**R**ltad), P-loop (gq**s**Glgk)[†], Switch I (ee**R**lPKtie)[†] |
| 4KV9, SmSEPT10•GDP (G dimer) | trans loop 1 (pt**GH**Slks) | α4 (lqkFka**R**ilse), P-loop (ge**T**Gigk)[†] |
| 4KVA, SmSEPT10•GTP (G dimer) | trans loop 1 (pt**GH**Slks) | α4 (lqkFka**R**ilse), P-loop (ge**T**gigk)[†] |
| 5AR1, ScCdc11•empty (quasi-G dimer)[‡] | trans loop 1 (ptg**H**Glke) | α4 (lklnkk**L**ime**D**) |

*Residues at positions corresponding to ScCdc3 Thr302 were identified based on structure-guided alignments. SEPT2, Arg198; SEPT3, Arg224; SEPT7, Gln210; SEPT9, Arg442; SEPT10, Arg199; Cdc11, Leu187.

[†]Residues from the other protomer across the G interface.

[‡]Dimer in solution is mediated by the C-terminal extension, and is unaffected by G interface mutations, thus contacts across the G interface are presumably crystal-induced.

of the trans loop 1 His may dictate which is ultimately chosen for incorporation, and Gdm could influence this choice.

In SEPT2•GppNHp, His158 is positioned in cis by contacts (~3 Å) with a Glu residue (Glu202) (*Sirajuddin et al., 2009*), which lies one turn away from Arg198 in the α4 helix (*Figure 4B*). The Gdm moiety of SEPT2 Arg198 (residue corresponding to Thr302 in Cdc3) is also within ~ 3 Å of the backbone amide carbonyl of His158 (*Figure 4B*, *Table 1*), and thus is equally well located to position His158. In CrSEPT and SEPT9•GTPγS, the Arg198 equivalent instead contacts the Glu202 equivalent (*Figure 4B*). Thus in some but not all septins the Gdm moiety of the α4 Arg is ideally positioned to contact the trans loop 1 His and thereby potentially bias partner choice during septin G dimerization. By contacting His262, Gdm bound near Thr302 in Cdc3 could bias Cdc3 towards homodimerization and away from heterodimerization with Cdc10.

To look for additional evidence in support of this model, we next asked which other residues are located in the vicinity (≤5 Å) of the trans loop 1 His in 13 available septin crystal structures. The α4 helix was the only region to meet this criterion in every structure, and, as expected, when Arg was present at the position corresponding to ScCdc3 Thr302, it was always within 5 Å (*Table 1*). Apart

from the α4 helix, the Switch II loop, and adjacent *trans* loop 1 residues, residues within 5 Å of the His fell within only two other regions: the P-loop and the Switch I loop (*Table 1*). These are precisely the regions wherein we found non-conservative G interface substitutions between fungal species that tolerate *CDC10* deletion and those that do not (*Figure 4B*). These observations provide strong support for the idea that Gdm binding between the ScCdc3 α4 helix and His262 mimics an ancestral evolutionary state and favors interaction with other similarly disposed Cdc3 molecules.

## The GTPase properties of the Cdc3 subunit dictate septin subunit composition

Our data support a model in which Gdm influences whether or not septin complexes incorporate a central homodimer of mutant Cdc10 molecules by influencing whether the Cdc3 *trans* loop 1 contacts the Switch II loop of another Cdc3 molecule, or points toward the site where nucleotide is normally bound by Cdc10. This model is reminiscent of the conclusions of a previous study, in which we proposed that the conformation of the Switch II loop in Cdc12 biases choice of its G-dimer partner (Cdc11 or Shs1)(*Weems and McMurray, 2017*). In that study, we showed that a Switch II mutation in Cdc12 is sufficient to bias partner choice. In an independent study (*Weems et al., 2014*), we found that a spontaneous mutation (D210G) in the equivalent Switch II residue in Cdc3 restores the ability of Cdc3 to interact with a nucleotide-free mutant Cdc10 at high temperatures. Like Gdm, the *cdc3(D210G)* mutation rescues the TS phenotype of *cdc10(D182N)* cells, but unlike Gdm it does so by restoring, rather than bypassing, incorporation of the mutant Cdc10 subunits; in other words, the Switch II mutation D210G biases Cdc3 partner choice towards nucleotide-free Cdc10, whereas Gdm biases partner choice away from it. Our model predicts that in *cdc3(D210G)* cells Gdm should be less able to exclude the mutant Cdc10 than in cells expressing WT Cdc3, because the mutant Cdc3 Switch II will be unable to accomplish the molecular contact(s) with the Cdc3 *trans* loop 1 that Gdm promotes. Indeed, Gdm slightly reduced, but did not eliminate, Cdc10(D182N)-GFP localization to septin rings in *cdc3(D210G)* cells (*Figure 8A*). These data are consistent with an important role for contacts between the Switch II loop and the *trans* loop 1 during Cdc3 homodimerization promoted by Gdm.

Since Cdc12 is an active GTPase, and the Switch II changes conformation upon GTP hydrolysis, we interpreted our previous results with Switch II-mutant Cdc12 as evidence that the Switch II conformation normally 'communicates' across the G interface the phosphorylation state of Cdc12's bound nucleotide (*Weems and McMurray, 2017*). We further bolstered this argument by mutating in Cdc12 a Thr residue in the Switch I loop that is required for septin GTPase activity (*Sirajuddin et al., 2009*; *Weems and McMurray, 2017*). Notably, Cdc3 lacks Thr in this position. How could Gdm operate via Switch II conformation on Cdc3, a 'dead' GTPase? We noticed in our phylogenetic analysis of Cdc3 homologs in other fungal species that many possess the 'catalytic Thr' (*Figure 4—figure supplement 1B*). In fact, there was a perfect correlation between the presence of the catalytic Thr (or Ser) and the α4 Arg whose Gdm group is presumably mimicked by Gdm in Cdc3 (*Figure 4—figure supplement 1B*). Analogous to our model of Cdc12 G-partner choice via slow GTP hydrolysis (*Weems and McMurray, 2017*), we reasoned that in a species with an active 'Cdc3' GTPase, a transient 'Cdc3'•GTP molecule might prefer to dimerize with 'Cdc10' in that species, and 'Cdc3'•GDP might prefer to form a homodimer, bypassing 'Cdc10' incorporation into septin complexes.

According to this model, because *Saccharomyces*, *Ashbya*, and *Kluyveromyces* species express catalytically dead Cdc3 molecules, only Cdc3•GTP exists in these cells, and only Cdc10-containing hetero-octamers are made. In *Aspergillus nidulans*, on the other hand, the Cdc3 homolog, AspB, possesses the catalytic Thr (*Figure 4—figure supplement 1B*, *Figure 8—figure supplement 1*), and this species is known to produce a mix of octamers containing AspD (the Cdc10 homolog) and AspD-less hexamers (*Hernández-Rodríguez et al., 2014*). We replaced the GTPase domain of ScCdc3 (residues 95–341) with that of AspB (residues 6–234; *Figure 8—figure supplement 1*) and assessed Cdc10 incorporation into complexes by measuring the ratio of Cdc10-mCherry to Shs1-GFP in septin rings of living cells. Consistent with our model, Cdc10-mCherry incorporation was markedly reduced compared to cells with WT Cdc3 (*Figure 8B*), indicating that the GTPase properties of AspB were sufficient to confer *Aspergillus*-like pathways of septin hetero-oligomer assembly, that is, frequent bypass of Cdc10 incorporation.

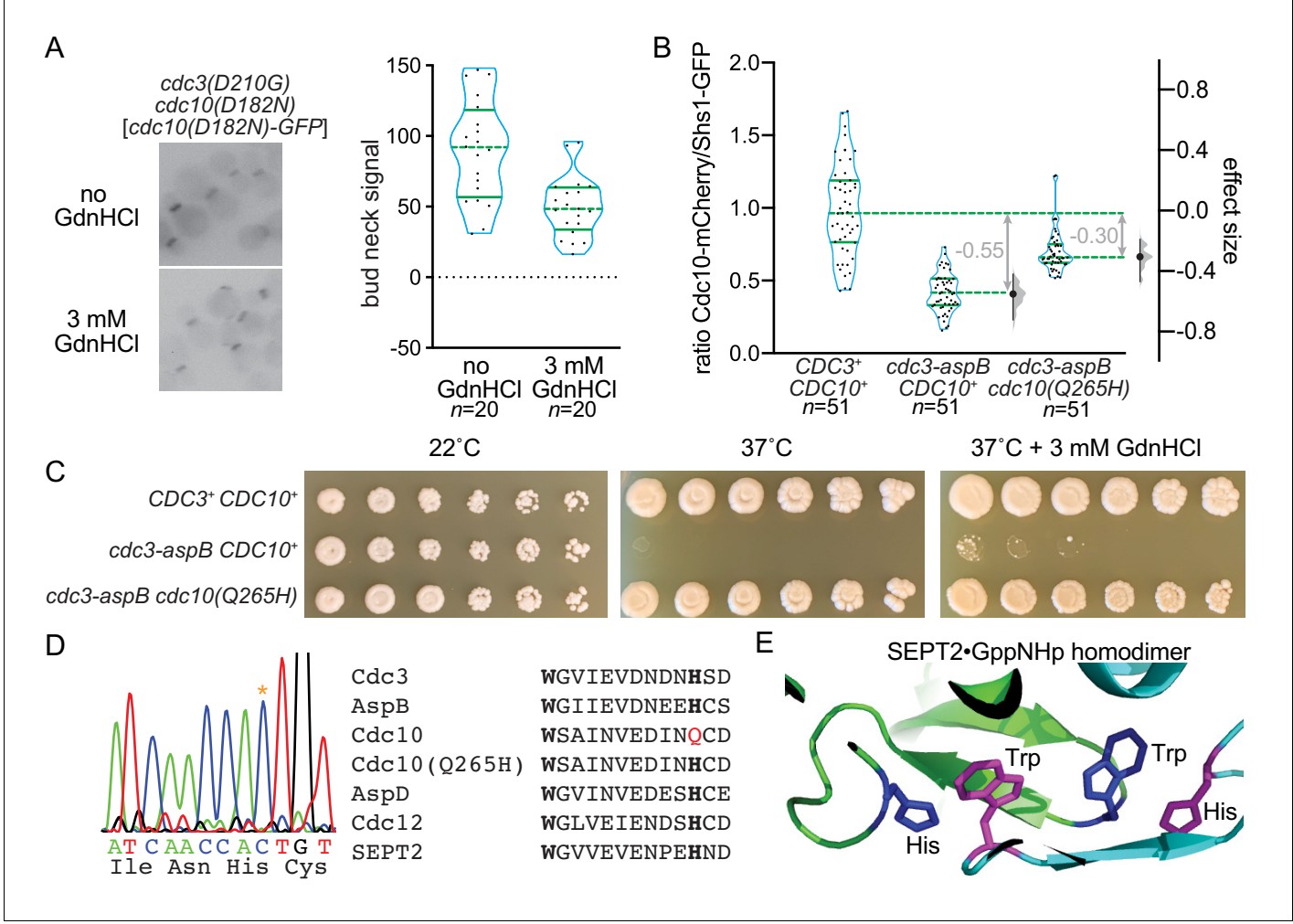

**Figure 8.** Reversing evolution to reactivate Cdc3 GTPase activity allows Cdc10 bypass during septin hetero-oligomerization. (**A**) Strain MMY0130, which carries the mutations *cdc3(D210G) cdc10(D182N)*, was transformed with plasmid YCpK-Cdc10-1-GFP and cultured in rich (YPD) medium containing G-418 for plasmid selection in the presence or absence of GdnHCl. Bud neck fluorescence was measured as in *Figure 2B*. (**B**) Haploid strains of the indicated genotypes expressing Cdc10-mCherry and Shs1-GFP were imaged as in (**A**) but in the absence of GdnHCl and for both mCherry and GFP fluorescence. For each cell, the ratio of mCherry to GFP signal was calculated; values are plotted on the left axis, with overlaid blue violin plots to show how the data are distributed, and median (dashed lines) and quartiles (solid lines) shown in green. The right axis shows the median difference ('effect size') for comparisons of the mutant strains against the shared wild-type control, plotted as bootstrap sampling distributions (gray violin plots). Each median difference is depicted as a block dot, with the value of that difference given in gray. Each 95% confidence interval is indicated by the ends of the vertical error bars. Strains were: MMY0342, MMY0343, and MMY0350. Protein sequence differences between the versions of Cdc3 present in these strains are detailed in *Figure 8—figure supplement 1*. Effects of specific individual substitutions on Cdc10-mCherry incorporation are shown in *Figure 8—figure supplement 2*. (**C**) Serial dilutions of the strains in (**B**) were spotted on rich (YPD) agar media with or without GdnHCl and incubated for 4 days at the indicated temperatures. (**D**) A portion of the *CDC10* coding sequence was amplified by PCR and sequenced. Asterisk in the chromatograph at left shows the mutation resulting in the amino acid change Q265H. Right, alignment of septin sequences surrounding the conserved Trp (bold) of the Sep4 motif and the His (bold) with which this Trp interacts across the G interface in the SEPT2 homodimer, as illustrated in (**E**). Red, Gln265 in wild-type Cdc10. (**E**) The Trp and His residues of the Sep4 motif in the SEPT2•GppNHp homodimer crystal structure (PDB 3FTQ). The online version of this article includes the following source data and figure supplement(s) for figure 8:

**Source data 1.** Cdc10(D182N)-GFP septin ring intensity values for *Figure 8A*.

**Source data 2.** Cdc10-mCherry/Shs1-GFP ratios for *Figure 8B*.

**Figure supplement 1.** Sequence of the Cdc3-AspB chimeric protein and comparison of key residues between Cdc3, AspB, and Cdc12.

**Figure supplement 2.** Five substitutions in the GTPase domain of Cdc3 are insufficient to promote bypass of Cdc10 during septin assembly.

**Figure supplement 2—source data 1.** Cdc10-mCherry/Shs1-GFP ratios for *Figure 8—figure supplement 2*.

*S. cerevisiae* cells expressing the Cdc3-AspB chimera as the only source of Cdc3 were unable to proliferate at 37°C (*Figure 8C*). We hypothesized that this defect reflects misfolding at high temperature of the Cdc3-AspB chimera to a conformation incapable of interacting with itself or with Cdc10, and searched for spontaneous suppressors of the TS phenotype. We obtained a suppressor in which proliferation at 37°C was restored (*Figure 8C*). Sequencing the coding region of Cdc10 revealed a single nucleotide change causing the amino acid substitution Q265H (*Figure 8D*); the Cdc3-AspB chimera was unchanged. In most septin structures, His in this position makes a critical contact across the G interface with a highly conserved Trp residue in the 'Sep4' motif (*Pan et al., 2007*) (*Figure 8E*), the same Trp we mutated in the *cdc3(W364A)* mutant (see *Figure 2D*). Since Gln replaces His here in ScCdc10 (and in the other fungal species with putatively GTPase-dead Cdc3 homologs; *Table 2*), the Cdc3•GTP–Cdc10•GDP interface must involve a different kind of interaction. We interpret these findings as evidence that stable association of Cdc10•GDP with GDP-bound Cdc3-AspB, rather than the Cdc3•GTP with which Cdc10 co-evolved, requires His–Trp contacts between Sep4 motifs of the sort found in other dimers between two GDP-bound septins. Indeed, incorporation of Cdc10-mCherry into septin rings was partially restored in *cdc3-aspB cdc10(Q265H)* cells (*Figure 8B*). Finally, Gdm was unable to fully rescue the TS phenotype of the *cdc3-aspB CDC10+* strain (*Figure 8C*), as expected if the site of Gdm action is occluded by the α4 Arg present in the Cdc3-AspB chimera. These data provide further support for the idea that during evolution ScCdc3 lost the ability to hydrolyze GTP to GDP and, consequently, the option of assembling septin complexes without a central Cdc10 homodimer.

The GTPase domains of Cdc3 and AspB are < 50% identical, and differ at many more positions than the five identified by our phylogenetic analysis as co-varying with the α4 Arg and the catalytic Thr (*Figure 8—figure supplement 1*). To ask if those five differences are sufficient to direct Cdc10 bypass during septin assembly, we used CRISPR-Cas9 to cut the endogenous *CDC3* coding sequence and, via homologous recombination, replace most of it with a 'recoded' gene encoding the same polypeptide sequence but using numerous synonymous codons, or with a similarly 'recoded' gene additionally encoding the five substitutions (P127E D128S K181T T302R Q306D). Recoding allowed us to obtain transformants in which recombination occurred at the ends of our donor templates, rather than in sequences immediately flanking the cut site, thus promoting incorporation of all five substitutions. We also obtained a transformant in which only two substitutions were incorporated, T302R Q306D, and we included this strain in our subsequent analysis. The recoded control strain and the *P127E D128S K181T T302R Q306D* and *T302R Q306D* mutant strains were indistinguishable from the parental *CDC3+* strain in terms of growth at all tested temperatures, and also with regard to the relative amounts of Cdc10-mCherry incorporated into septin rings (*Figure 8—figure supplement 2*). Thus, although we cannot rule out the possibility that recoding with synonymous codons alters Cdc3 co-translational folding in some way that masks effects of the amino

**Table 2.** Co-variation among fungal species of the "catalytic Thr" in Cdc3 homologs and His in the Cdc10 trans loop 1.

| Species | Amino acid at position corresponding to ScCdc3 Lys181 | Amino acid at position corresponding to ScCdc10 Lys155 | Amino acid at position corresponding to ScCdc10 Gln265 |
|---|---|---|---|
| *S. cerevisiae* | Lys | Lys | Gln |
| *A. gossypii* | Lys | Lys | Gln |
| *K. lactis* | Lys | Lys | Gln |
| *C. albicans* | Ser | Gln | Gln |
| *A. nidulans* | Thr | His | His |
| *C. cinerea* | Thr | His | His |
| *U. maydis* | Thr | His | His |
| *N. crassa* | Thr | His | His |
| *M. oryzae* | Thr | His | His |
| *F. graminearum* | Thr | His | His |
| *C. neoformans* | Thr | His | His |
| *S. pombe* | Thr | His | His |

acid changes we introduced, it appears that additional sequence changes are needed to 'reverse evolution' and restore the ability of Cdc3 to homodimerize robustly in vivo.

## Discussion

### Septin G dimer partner selection by 'tuning' the *trans* loop 1 His via α4 Arg or Gdm

We propose that Gdm binding adjacent to ScCdc3 Thr302 provides molecular contacts that position His262 to interact in trans with an Asp residue from a septin Switch II loop in a 'GTP state', such as that provided by another Cdc3•GTP. The spontaneous mutation we previously found to promote Cdc3 homodimerization (*McMurray et al., 2011*) alters a *trans* loop 1 residue immediately adjacent to Cdc3 His262, Gly261. Val in this position probably also 'tunes' the configuration of His262 to favor Cdc3 homodimerization via interaction with Cdc3 Asp201. Specific configurations of the Switch II loop clearly dictate G dimer partner choice, as mutating the residue corresponding to Cdc3 Asp210 promotes both stable Cdc3 heterodimerization with nucleotide-free Cdc10 (*Weems et al., 2014*) and biased selection of Shs1 by Cdc12 (*Weems and McMurray, 2017*). Mutating a nearby Asp in the Switch II loop of human SEPT7 also allows homodimerization when only a non-hydrolyzable GTP analog is available (*Zent and Wittinghofer, 2014*). 'Tuning' by Gdm via *cis* interactions presumably occurs prior to G dimerization and is likely a transient event: once the *trans* loop 1 His has engaged either the Switch II Asp or nucleotide (or nearby where nucleotide normally binds, in the case of nucleotide-free mutant Cdc10), the Arg–His interaction probably need not persist post-dimerization. It follows that Gdm may not remain bound to Cdc3 following G dimerization, and thus falls under a broad definition of a 'pharmacological' or 'chemical' chaperone.

Since in otherwise WT cells Gdm promotes bypass of mutant Cdc10 molecules with substitutions in various locations in the G interface and nucleotide-binding pocket, but not WT Cdc10, additional contacts between WT Cdc10•GDP and Cdc3•GTP likely favor Cdc3–Cdc10 heterodimerization over Cdc3 homodimerization despite the presence of Gdm. Poorer GdnHCl rescue of *cdc10(G44D)* (P-loop mutant) relative to *cdc10(D182N)* (G4 mutant) and *cdc10(G100E)* (Switch II mutant) may reflect a comparatively better ability of Cdc10(G44D) molecules to, despite temperature-induced misfolding, occupy the Cdc3 G interface and resist exclusion, thereby interfering with hexamer assembly. Bound nucleotide itself may provide the key contacts that distinguish WT Cdc10 from the mutants, with Cdc10(G44D) subunits being more successful than the other mutants at binding nucleotide. At moderate temperatures (22–27°C) Cdc3 molecules carrying the G365R substitution are slow to acquire the conformation competent for interaction with Cdc10 (*Schaefer et al., 2016*). Our observation that GdnHCl exacerbated the TS phenotypes of *cdc3(G365R)* cells (*Figure 1A*) thus provides further evidence of a bias imposed by Gdm on Cdc3 toward conformations that are suboptimal for interaction with Cdc10, the effects of which are masked in WT cells but become obvious when combined with additional folding biases like the G365R substitution.

Whereas Gdm was unable to drive Cdc10 bypass in WT cells, swapping the GTPase domain of Cdc3 for that of AspB was sufficient to promote bypass of WT Cdc10 (*Figure 8B*), consistent with the idea that Cdc3•Gdm only partially mimics the ancestral homodimer configuration. Introducing into Cdc3 the five residues (including the catalytic Thr and α4 Arg) that are found in Cdc3 homologs in species that tolerate loss of the Cdc10 homolog was unable to drive exclusion of Cdc10 from septin complexes (*Figure 8—figure supplement 2*). Cdc12 natively possesses four of these five residues (*Figure 8—figure supplement 1*), yet forcing septin hexamer assembly via non-native Cdc12 homodimers (by deleting *SHS1* and *CDC11*) results in *S. cerevisiae* cells that, like *cdc10Δ*, cannot proliferate at 37°C (*McMurray et al., 2011*). Clearly GTPase activity and the presence of these residues is insufficient for robust septin homodimerization, and additional changes are likely key.

Other factors promote Cdc3 homodimerization in similar ways as GdnHCl, as illustrated by the collective ability of the G261V mutation, excess GTP and a synthetic lipid monolayer to drive filament formation by Cdc10-less complexes in vitro (*Bertin et al., 2010*). This effect requires high protein concentration (0.15 mg/mL) (*Bertin et al., 2010*). Even higher protein concentration (1 mg/mL) bypasses the need for Cdc3 mutations and some extrinsic cofactors, though excess GTP is still critical (*Farkasovsky et al., 2005*). These observations support a 'conformational ensemble' model of Cdc3 folding, in which at any given time a small proportion of Cdc3 molecules adopt the

homodimerization-competent conformation. While this proportion can be increased in various ways, another way to populate even a very rare conformation is to simply increase the total number of Cdc3 molecules.

## Insights into major events during septin evolution

Although septins are absent from land plants, protein transport across the outer envelope membrane of chloroplasts is mediated by TOC complexes containing a small GTPase subunit, Toc34, that shares high structural homology to septins. Others have noted (*Weirich et al., 2008*) that the septin *trans* loop 1 is in a similar position to a Toc34 interface loop that makes key contacts with nucleotide bound by the partner protein across a homodimer interface (*Figure 9A*). In both pea (*Pisum sativum*) and *Arabidopsis thaliana* an Arg residue in the Toc α4 helix projects towards the homodimer interface, to within 5 Å of the *trans* loop 1 equivalent (*Figure 9A*). Arg in this position is conserved among all Toc34 homologs (*Figure 9B*). We conclude that Arg-mediated α4 helix tuning of the *trans* loop 1 for specific dimerization events across the nucleotide-binding-pocket-containing interface is an evolutionarily ancient mechanism that predated the appearance of septins per se.

Following loss of the α4 Arg in the fungal lineage giving rise to *S. cerevisiae*, *A. gossypii*, and *K. lactis*, the amino acid at the position equivalent to ScCdc3 Thr302 diverged rapidly. Ser or Ala appear in its place in other species within the genus *Saccharomyces*, which share >86% sequence identity with ScCdc3 (*Figure 9C*). By comparison, within the genus *Candida* Arg is invariant among six species sharing as little as 35% identity with *C. albicans* Cdc3 (*Figure 9C*). These observations are consistent with the notion that Thr in this position does not make specific structural contributions to septin assembly.

One clear prediction from this model is that in the two fungal lineages distinguished by the presence or absence of the Cdc3 'catalytic' Thr, the Cdc10 subunit should have evolved differently, to recognize in one case only Cdc3•GTP and, in the other, either Cdc3•GTP and Cdc3•GDP. Our unbiased identification of the *cdc10(Q265H)* mutation in cells expressing the Cdc3-AspB chimera provided one example of how Cdc10 adapted to interacting with a Cdc3 subunit 'fixed' in the GTP-bound state. Others previously noted (*Sirajuddin et al., 2009*) that ScCdc10 is unusual among septins in that it has Lys rather than His in the *trans* loop 1. If the residue in this position acts to promote interaction with a G dimer partner in a specific nucleotide state, this may explain why Lys in this position is shared by AgCdc10 and KlCdc10, but not any of the Cdc10 homologs in the fungal species in which the Cdc3 homolog retains the 'catalytic Thr' (*Table 2*). Co-variation of these residues likely reflects co-evolution. *Figure 9—figure supplement 1A* illustrates a possible pathway of such changes during septin evolution.

Were Cdc10-less hexamers specifically selected against during evolution of a fungal lineage? Compared to other closely-related fungi, *Ashbya*, *Kluyveromyces*, and *Saccharomyces* share little in common with regard to the cellular functions in which septins are known to operate, such as polarity, morphogenesis, cytokinesis, or sporulation. Hence it is hard to imagine how losing septin hexamers could provide a specific selective advantage to a common ancestor, though this may reflect our incomplete understanding of what septin hexamers do differently than septin octamers. On the other hand, the assembly of hexamers liberates the Cdc10 subunits to act elsewhere, and specific functions of 'free' Cdc10 homologs could represent an evolutionary pressure point to maintain GTPase activity in Cdc3 homologs. Indeed, in *U. maydis* (*Alvarez-Tabarés and Pérez-Martín, 2010*) and *A. nidulans* (*Hernández-Rodríguez et al., 2014*) the Cdc10 homologs are found in distinct cellular structures independently of the other septins, where they may function independently, as well.

Many human cell types also natively produce both septin hexamers and octamers (*Abbey et al., 2016*; *Kim et al., 2011*; *Sellin et al., 2014*; *Sellin et al., 2011a*; *Sellin et al., 2011b*). Incorporation of a subunit from the SEPT3 group (SEPT3, SEPT9, or SEPT12) appears to confer functions to human octamers that hexamers lack, such as microtubule bundling (*Bai et al., 2013*), vesicle trafficking in neurons (*Karasmanis et al., 2018*), directing proper spermatogenesis (*Kuo et al., 2015*), and membrane abscission at the end of cytokinesis (*Estey et al., 2010*). Human and yeast septin octamers were thought to differ in where within the complex the 'extra' subunits incorporate, with SEPT3-group septins like SEPT9 occupying the 'terminal' positions at both octamer ends (*Kim et al., 2011*). However, two recent reports (*Mendonça et al., 2019*; *Soroor et al., 2019*) suggest that in fact SEPT3-group septins form a central homodimer flanked by SEPT7, with SEPT6 and SEPT2 occupying the penultimate and terminal positions, respectively (*McMurray and Thorner, 2019*). In this 'revised'

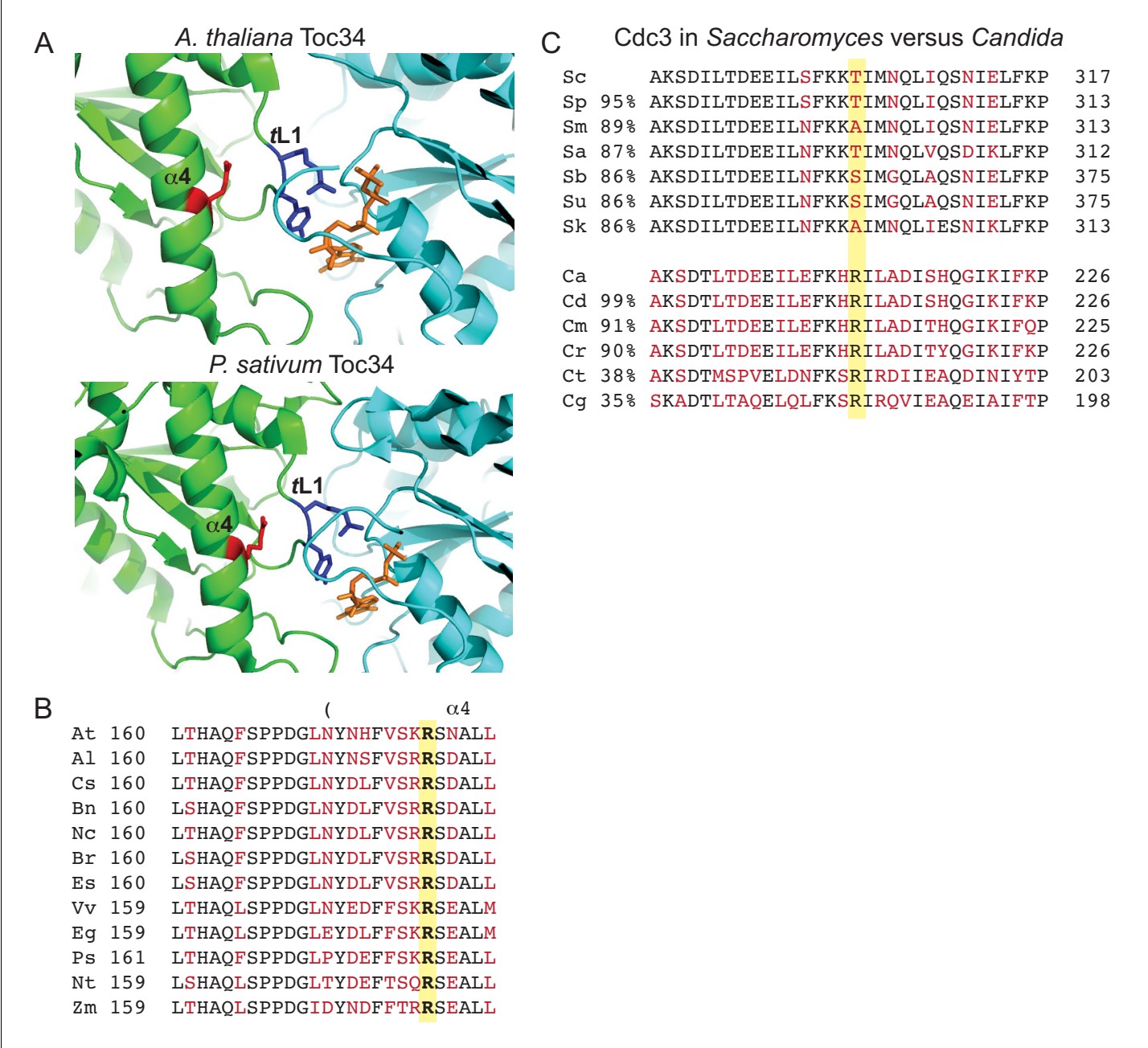

**Figure 9.** Conservation of contacts between α4 Arg and *trans* loop 1 in the distant septin relative Toc34 and among *Candida* species, and drift at the α4 Arg position in *Saccharomyces* species. (**A**) Structures of homodimer interface of Toc34 from *Arabidopsis thaliana* (GppNHp-bound, PDB 3bb3, top) and *Pisum sativum* (GDP-bound, PDB 1h65, bottom) with nucleotide in orange, α4 Arg in red and nucleotide-contacting *trans* loop 1 ('TL1') residues in blue. (**B**) Sequence alignment of a region in and around the α4 helix of Toc34 homologs in various plant species, with identical and variable amino acids in black and red, respectively, and the α4 Arg in bold. *A. thaliania*, At (NP_196119); *A. lyrata*, Al (XP_002873181); *Camelina sativa*, Cs (XP_010423506); *Brassica napus*, Bn (CDX98810); *Noccaea caerulescens*, Nc (JAU73517); *B. rapa*, Br (XP_009130855); *Eutrema salsugineum*, Es (XP_006398957); *Vitis vinifera*, Vv (XP_002274573); *Eucalyptus grandis*, Eg (XP_010063039); *P. sativum*, Ps (Q41009); *Nicotiana tabacum*, Nt (XP_016456518); *Zea mays*, Zm (ACG34570). (**C**) Sequence alignments of a region surrounding ScCdc3 Thr302 (highlighted in yellow), with identical and variable amino acids in black and red, respectively. Percent sequence identity over the entirety of Cdc3, relative to ScCdc3 or CaCdc3, is given at left. Alignments of *Saccharomyces* species were performed using the Saccharomyces Genome Database (http://www.yeastgenome.org). Sc, *S. cerevisiae*; Sp, *S. paradoxus*; Sm, *S. mikatae*; Sa, *S. aboricola*; Sb, *S. bayanus*; Su, *S. uvarum*; Sk, *S. kudriavzevii. Candida* alignments were performed at the NCBI Blast server. Ca, *C. albicans* (EEQ42751); Cd, *C. dubliniensis* (CAX44656); Cm, *C. maltosa* (EMG46918); Cr, *C. tropicalis* (XP_002550438); Ct, *C. tenuis* (EGV60285); Cg, *C. glabrata* (CAG57768). *Figure 9—figure supplement 1* presents a speculative model for how the position corresponding to ScCdc3 Thr302 and other relevant molecular features may have changed during septin evolution.

*Figure 9 continued on next page*

*Figure 9 continued*

The online version of this article includes the following figure supplement(s) for figure 9:

**Figure supplement 1.** Model for the roles of key molecular features during the evolution of septin complexes.

organization, the GTPase-dead subunit position (the SEPT6 group) corresponds to that of Cdc12, and the Cdc3 position is occupied by SEPT7, an active GTPase (see *Figure 9—figure supplement 1B*). Notably, a mutation (S63N) that doubles SEPT7 GTPase activity in vitro shifts assembly in vivo from primarily hetero-octamers to smaller complexes (*Abbey et al., 2016*), consistent with our model that septins in the GDP-bound state are prone to homodimerization across the G interface and exclusion of intervening subunits.

SEPT9 and SEPT3 are active GTPases and, like Cdc10 (*Versele and Thorner, 2004*), as monomers they hydrolyze GTP faster than other septins (*Macedo et al., 2013*; *Zent and Wittinghofer, 2014*). We previously proposed that rapid GTP hydrolysis by monomeric Cdc10 allows it to homodimerize prior to interacting with Cdc3, such that Cdc10•GDP encounters Cdc3•GTP during hetero-oligomerization (*Weems and McMurray, 2017*). If human septins assemble similarly, then hexamers lacking SEPT3-family subunits are not made by eviction of central homodimers from pre-existing hetero-octamers. Instead, the outcome of hexamer or octamer is decided upon de novo assembly and depends upon the phosphorylation state of the nucleotide bound by SEPT7. Indeed, induced over-expression of tagged SEPT3-family septins in human K562 cells demonstrated that preexisting hexamers gradually disappeared by dilution during cell division and were replaced by newly-made octamers (*Sellin et al., 2014*). In that study, the 'excess' SEPT9 persisted for days (*Sellin et al., 2014*). Given subsequent results pointing to a role for 'free' SEPT9 in vesicle trafficking in mouse hippocampal neurons (*Karasmanis et al., 2018*), in specific cellular circumstances slow SEPT7 GTP hydrolysis may be crucial for generating 'free' SEPT3-family septins that perform independent functions.

Cdc10 and SEPT9 fall into one ancient evolutionary clade, whereas Cdc3, Cdc12, SEPT6 and SEPT7 all appear to have distinct origins (*Auxier et al., 2019*). Thus the loss of septin GTPase activity occurred multiple times independently. By demonstrating that GTPase activity facilitates promiscuity during dimerization, and thereby expands the repertoire of septin hetero-oligomers, our findings provide the first molecular rationale for the evolution of GTPase-dead septins: to narrow the variety of building blocks available for higher-order assemblies. A similar logic may explain the most famous example of loss of GTPase activity of a cytoskeletal protein, that of the α subunit of the tubulin heterodimer.

## Chemical rescue by guanidine hydrochloride in vivo

The ability of Gdm to occupy vacancies created by the mutation of Arg residues is well documented in vitro for a wide variety of enzymes (*Baldwin et al., 1998*; *Barnett et al., 2010*; *Boehlein et al., 1997*; *Dugdale et al., 2010*; *Goedl and Nidetzky, 2008*; *Guillén Schlippe and Hedstrom, 2005*; *Hung et al., 2014*; *Lehoux and Mitra, 2000*; *Phillips et al., 1992*; *Rynkiewicz and Seaton, 1996*), but effects of GdnHCl on living cells were limited to Gdm binding at sites in WT proteins or nucleic acids: in the ATP-binding pocket of Hsp104 and its relatives (*Zeymer et al., 2013*), in the intracellular pore of voltage-gated potassium channels (*Kalia and Swartz, 2011*), in the enteroviral 2C protein (*Sadeghipour et al., 2012*), and, most recently, in bacterial riboswitches that sense intracellular Gdm and induce a cellular response to detoxify it (*Nelson et al., 2017*). The function of a rationally-designed Arg-mutant kinase can be rescued in vitro by GdnHCl (*Williams et al., 2000*), and the same mutant is rescued by imidazole in vivo (*Qiao et al., 2006*), but our findings that GdnHCl can functionally replace an Arg in Cdc3 represent the first documented case of 'chemical rescue' by GdnHCl in vivo.

The apparent requirement for GdnHCl to be present during Cdc3 folding illustrates how our case of GdnHCl mimicry of a 'missing' Arg is fundamentally distinct from the other examples of 'chemical rescue' by GdnHCl. In the other cases, a protein that had evolved to adopt a specific conformation was mutated at a single Arg in order to perturb this conformation (or, in many cases, interaction with a substrate molecule), and Gdm was able to restore the native fold/interaction. In those prior studies, the mutated protein was already near the native conformation and only a small effect of

GdnHCl was required to achieve the native conformation. In our case, a WT protein that had evolved to avoid a particular conformation was induced to reach that non-native conformation only when GdnHCl was present during its folding. During Cdc3 folding in the absence of Gdm, other intramolecular interactions drive acquisition of the low-energy native conformation that disfavors homodimerization. Gdm would have to act early, in a manner equivalent to how canonical chaperones direct the folding pathways of their clients. The ability of the T302R mutation to block, but not to mimic, the effect of GdnHCl could be explained by an ability of Gdm to make intramolecular contacts within Cdc3 that, because it is tethered to the polypeptide backbone, the Gdm moiety of an Arg side chain cannot. Our findings do not exclude the possibility that additional Gdm molecules bind elsewhere in Cdc3 and contribute to Cdc3 folding, with occupancy of the site near Thr302 being necessary but not sufficient.

In yeast cells cultured in 1 mM GdnHCl the intracellular concentration is ~ 20 mM (*Jones et al., 2003*). Thus in the conditions in which we observed *cdc10* rescue Gdm likely binds to hundreds or thousands of proteins, with few adverse effects. Indeed, GdnHCl is approved by the Food and Drug Administration of the USA for use in humans, and at low doses has few serious side effects (*Oh et al., 1997*). Exploiting the ability of Gdm to replace Arg in living cells represents a compelling area for future research, particularly as a possible therapeutic 'pharmacoperone' (*Conn et al., 2014*). Finally, given the evidence that bacteria evolved ways to manage intracellular Gdm (including actively exporting it [*Kermani et al., 2018*]), Gdm may, like the naturally occurring osmolyte trimethylamine N-oxide (*Bandyopadhyay et al., 2012*) and the chaperone Hsp90 (*Lindquist, 2009*), influence evolution by buffering against the phenotypic consequences of mutations.

## Materials and methods

### Fungal strains and plasmids

Yeast were transformed using the Frozen-EZ Yeast Transformation II Kit (Zymo Research). Genetic manipulations were otherwise performed according to standard methods (*Amberg et al., 2005*), except for the creation of the *cdc3-aspB* chimera and the *CDC3(P127E D128S K181T T302R Q306D)* and *CDC3(T302R Q306D)* strains and associated 'recoded' control strain, which were made in yeast strain JTY5397 using CRISPR-Cas9 cleavage of the *CDC3* locus and repair with PCR products as donor templates, following an established protocol (*Akhmetov et al., 2018*) and using plasmid pEM-CDC3-CRISPR1, a gift of Ed Marcotte, which encodes Cas9 and a guide RNA targeting nucleotides 1039–1069 of the *CDC3* ORF (5' GATATTGTAGAGAACTACAG 3'). To create the donor template for the *cdc3-aspB* chimera, a portion of the *aspB* coding sequence from the *aspB* plasmid pRL10, which was a gift of Michelle Momany and is based on the yeast two-hybrid vector pGBKT7 (Takara Bio/Clontech), was used as template for a PCR reaction with Q5 polymerase (New England Biolabs) and primers Cdc3AspBfw and Cdc3AspBextend1 according to the polymerase manufacturer's instructions. The resulting product, which included the sequence encoding part of AspB flanked by *CDC3* sequences, was used as template for a second Q5 reaction with primers Cdc3AspBfw and Cdc3GTPase_extend2, in order to extend the *CDC3* homology to span the site of the Cas9 cut. To create the donor template for the *CDC3(P127E D128S K181T T302R Q306D)* strain and its 'recoded' control, an 870-nucleotide segment of the Cdc3 ORF corresponding to the GTPase domain was synthesized (Integrated DNA Technologies) with multiple synonymous codons replacing the native sequence, representing 67 nucleotide changes but no amino acid substitutions. An otherwise identical sequence also including the *P127E D128S K181T T302R Q306D* mutations was also synthesized. Donor template PCRs were done with Q5 and primers G1KTTRQDfw and Cdc3recodere. Transformants were screened by PCR of the *CDC3* locus and subsequent sequencing.

### Media and additives

*S. cerevisiae* media: Rich growth medium was YPD (1% yeast extract (#Y20020, Research Products International Corp., Mount Prospect, IL), 2% peptone (#P20241, RPI Corp.), 2% dextrose (#G32045, RPI Corp.)). Synthetic growth medium was based on YC (0.1 g/L Arg, Leu, Lys, Thr, Trp and uracil; 0.05 g/L Asp, His, Ile, Met, Phe, Pro, Ser, Tyr and Val; 0.01 g/L adenine; 1.7 g/L Yeast Nitrogen Base (YNB) without amino acids or ammonium sulfate; 5 g/L ammonium sulfate; 2% dextrose) with individual components (from Sigma Aldrich, St. Louis, MO, or RPI Corp.) eliminated as appropriate for

plasmid selection. For solid media, agar (#A20030, RPI Corp.) was added to 2%. For counterselection against *URA3*, 5-fluoro-orotic acid monohydrate (#F5050, United States Biological, Salem, MA) was added to modified YC medium (1 g/L Pro in place of ammonium sulfate, 0.02 g/L uracil) to final 0.6 g/L. For counterselection against *LYS2*, α-aminoadipate (#A1374-09, US Biological) at 2 g/L replaced ammonium sulfate in YC medium containing only YNB, uracil, His, Leu, Lys, Met, Trp, and dextrose. G-418 sulfate (Geneticin, #G1000, US Biological) was added to YPD at 200 µg/mL for selection of *kanMX*.

A. gossypii media: Ashbya strains were grown in 10 mL Ashbya Full Medium (AFM, 1% casein peptone, 1% yeast casein extract, 2% dextrose and 0.1% myo-inositol) containing ampicillin (100 µg/mL), CloNAT (50 µg/mL) and G-418 (200 µg/mL). For imaging, cells were washed into 2X low-fluorescence minimal medium (per liter: 3.4 g YNB without amino acids or ammonium sulfate, 2 g Asp, 2 g myo-inositol, 40 g dextrose, 20 mg adenine, 21 g 3-(N-morpholino)propanesulfonic acid (MOPS), pH adjusted to 7.0 with sodium hydroxide).

GdnHCl (#G4505, Sigma Aldrich, St. Louis, MO), ArgHCl (#A5131, Sigma Aldrich), urea (Bio-Rad # 1610730), aminoguanidine hydrochloride (#sc-202931, Santa Cruz Biotechnology, Santa Cruz, CA) and N-ethylguanidine hydrochloride (#sc-269833, Santa Cruz Biotechnology) were dissolved in water.

## Growth curves

Growth curves were generated using a Cytation 3 plate reader (BioTek, Winooski, Vermont) as described previously (*Schaefer et al., 2016*) using a starting overnight culture of the *cdc10(D182N)* strain CBY06417 grown at 22°C prior to dilution into 200 µL YPD medium in each well.

## Immunofluorescence

Immunofluoresence was performed as described previously (*Pringle et al., 1989*) using an anti-Cdc11 primary antibody (#sc7170, Santa Cruz Biotechnology, Santa Cruz, CA) and Alexa-Fluor 488-labeled anti-rabbit secondary antibody (#20012, Biotium, Inc, Fremont, CA) at a 1:5000 and 1:1000 dilution, respectively.

## Bacterial expression and purification of septins

BL21 Star(DE3) *E. coli* cells (ThermoFisher Scientific # C601003) were transformed with plasmids pMVB121 (encoding 6xHis-Cdc12) and pMVB133 (encoding untagged Cdc3 and Cdc11) and grown overnight in 3 mL LB medium (per L: 10 g tryptone, 5 g yeast extract, 10 g NaCl) with ampicillin (50 µg/mL), chloramphenicol (34 µg/mL), and varying amounts of GdnHCl (from 0 mM to 96 mM) at 37°C in glass culture tubes with agitation. Growth was assessed qualitatively and was equivalent for all cultures, so 50 mM was chosen as the concentration for subsequent experiments. 1 L 2XTY (per L: 16 g Tryptone, 10 g yeast extract, 5 g NaCl) cultures with ampicillin, chloramphenicol and with or without 50 mM GdnHCl were grown at 37°C with shaking to $OD_{600}$ between 0.6 and 1.0, at which point IPTG was added to 0.1 mM and the culture was incubated overnight. Cells were collected by centrifugation, resuspended in a minimal volume of lysis buffer (300 mM NaCl, 2 mM $MgCl_2$, 40 µM GTP, 1 mM EDTA, 5 mM β-mercaptoethanol, 0.5% Tween-20, 12% glycerol, 50 mM Tris-HCl, pH 8.0, with or without 50 mM GdnHCl) and frozen by dripping into liquid nitrogen. Cells were thawed in 30 mL lysis buffer containing protease inhibitors (Complete EDTA-free, Roche #11836170001), lysozyme and benzonase (Sigma-Aldrich #E8263). Cells were sonicated for 5 min total (30 s on, 30 s off) with a tip sonicator. Lysates were clarified by centrifugation at 12,000xg at 4°C and mixed with 200 µL of a slurry of $Ni^{2+}$-NTA agarose (Qiagen #30210) that had been equilibrated in lysis buffer. After rotating for 1 hr at 4°C, beads were collected by centrifugation, washed two times with 5 mL of 300 mM NaCl, 20 mM imidazole, 0.1% Tween-20, and 50 mM Tris-HCl, pH 8.0, with or without 50 mM GdnHCl. The beads were poured into a column and the septin complex was eluted in fractions from the beads with 500 mM NaCl, 40 µM GTP, 500 mM imidazole, 50 mM Tris-HCl, pH 8.0, with or without 50 mM GdnHCl. Aliquots of fractions were analyzed by SDS-PAGE and Coomassie staining, and imidazole was removed from the fraction containing the peak of septin complex via buffer exchange into 500 mM NaCl, 40 µM GTP, 50 mM Tris-HCl, pH 8.0 using a PD-10 column (GE Healthcare LifeSciences #17085101). The sample was further purified by size exclusion chromatography using a Superdex 200 column and 50 mM Tris-HCl, 300 mM NaCl pH 8, supplemented with or

without 50 mM GdnHCl). The fractions were analyzed by SDS-PAGE and Coomassie staining to identify samples for electron microscopy.

## Electron microscopy

For EM of yeast cells, cells were cultured at 37˚C in YPD medium with 3 mM GdnHCl and collected by vacuum filtration, then vitrified by high pressure freezing using a HPM100 (Leica Microsystems, Wetzlar, Germany) apparatus. The frozen cells were cryo-substituted using a Leica ASF2 device in a medium containing osmium tetroxide (1%), water (5%) and uranyl acetate (0.1%) in acetone using a protocol described elsewhere (*Bertin and Nogales, 2016*). The cells were embedded into Epon before being sectioned into 50 nm sections using an ultramicrotome UC6 (Leica) equipped with a 4.5 mm diamond knife (Diatome, Hatfield, PA). The resulting sections were deposited onto copper electron microscopy grids (mesh size of 100). The images were collected with a 120 kV Lab6 microscope (Technai Spirit, FEI, Eindhoven, Netherlands) equipped with a CCD Quemesa camera (Olympus, Tokyo, Japan).

For EM of septin complexes purified from *E. coli*, 4 µL of sample at a final concentration of 0.01 mg/mL for high salt conditions or 0.1 mg/mL for low salt conditions were absorbed for 30 s on a glow-discharged carbon coated grid (Electron Microscopy Sciences, ref CF300-CU). The grids were then negatively stained for 1 min using 2% uranyl formate. Data was collected using either a Tecnai Spirit microscope (Thermofischer, FEI, Eindhoven, The Netherlands) operated at an acceleration voltage of 80Kv and equipped with a Quemesa (Olympus) camera or with a Lab6 G2 Tecnai (Thermo-Fisher, FEI, Eindhoven, the Netherlands) operated at an acceleration voltage of 200 kV. The data was acquired using a 4k × 4k F416 CMOS camera (TVIPS) in an automated manner using the EMTools software suite (TVIPS).

## Two dimensional image processing for single-particle EM images

Square boxes were hand-picked from the images respectively using the boxer tool from the EMAN software suite (*Ludtke et al., 1999*). 4233 boxes of 135 pixels were picked from the control sample while 3247 boxes of 203 pixels were picked from images collected of the sample grown in the presence of GdnHCl. Subsequent processing was carried out using SPIDER (*Frank et al., 1996*). After normalization of the particles, a non-biased reference-free algorithm was used to generate 20 classes. Multi-reference alignment followed by classification was carried out to generate about 50 classes. Fast Fourier transform analysis and power spectrum generation were performed in ImageJ (*Schneider et al., 2012*).

## Fluorescence microscopy

All images of *S. cerevisiae* were captured using an EVOSfl all-in-one microscope (Advanced Microscopy Group, Mill Creek, Washington) using a 60X objective and Texas Red, RFP, or GFP LED/filter cubes, as described previously (*Johnson et al., 2015*). Bud neck fluorescence was quantified using line scans as described previously (*Johnson et al., 2015*; *Weems and McMurray, 2017*). We developed a new macro for ImageJ/FIJI, called 'Get plot profile Min-Max', to facilitate such quantification. Intensity values or ratios thereof were plotted using GraphPad Prism 8.0, using the medium smoothing (kernel density) setting for violin plots. When necessary for presentation, images were inverted and brightness- and contrast-adjusted using Adobe Photoshop (Adobe Systems Incorporated, San Jose, California) or ImageJ (*Schneider et al., 2012*), always the same way for every image of the same type.

*Ashbya* cells in minimal low-fluorescence medium were mounted onto 2% agarose gel pads and the edges were sealed with Valap (a 1:1:1 mixture of vaseline, lanolin, and paraffin). Images were acquired using a Zeiss Axioimager-M1 upright light microscope (Carl Zeiss, Jenna, Germany) equipped with a Plan-Apochromat 63X/1.4 numerical aperture oil objective and an Exfo X-Cite 120 lamp. Fluorescence imaging was performed using Zeiss 38HE filter cubes (GFP). Images were acquired using an Orca-AG charge-coupled device (CCD) camera driven by µManager.

## In silico docking, structure prediction, and analysis

Visualization and rendering of structures in PDB format was performed using MacPyMol (Schrödinger, New York, NY). Residues within 5 Å of other residues were identified using the 'around'

command. Structure prediction for ScCdc3 was performed using the I-TASSER server (http://zhan-glab.ccmb.med.umich.edu/I-TASSER/) (*Zhang, 2008*) and ScCdc3 sequence truncated to amino acids 116–411 to match the template model, SEPT2•GppNHp (PDB 3FTQ). Molecular docking was performed using a local installation of AutoDock Vina (*Trott and Olson, 2010*) via Chimera 1.11 (http://www.rbvi.ucsf.edu/chimera) (*Pettersen et al., 2004*), using a search volume containing the entire 'receptor' model.

## Phylogenetic tree creation

The fungal protein sequences listed in the legend to *Figure 4—figure supplement 1* were analyzed using the Phylogeny.fr platform online (http://www.phylogeny.fr/) (*Dereeper et al., 2008*) to generate a phylogenetic tree using the 'One Click' method with default settings.

## Sequence alignments

Multiple alignments were performed using the COBALT tool at the NCBI server (https://www.ncbi.nlm.nih.gov/tools/cobalt/) or, for sequences of yeast species not available via NCBI, using the 'Fungal Alignment' function at the Yeast Genome Database (*Saccharomyces* Genome Database, https://www.yeastgenome.org/). In some cases, sequences obtained via the Yeast Genome Database were aligned with other sequences via COBALT.

## DNA sequencing

Genomic DNA from the *cdc3-AspB* suppressor strain was isolated as described previously and used as template in Q5 PCR reactions with primers 5'cdc3fw and 3'cdc3re, or 5'cdc10fw and 3'cdc10re performed according to the polymerase manufacturer's instructions. Following treatment with alkaline phosphatase and Exonuclease I (Thermo Scientific Fermentas), the purified PCR product was directly sequenced at the sequencing facility of the Barbara Davis Center for Childhood Obesity, with the same primers used for amplification and, for *CDC10*, with primer cdc10midfw. To confirm correct strain construction, sequencing of other loci was performed in an equivalent manner with appropriate primers (S2 Table).

## Choice of sample sizes and statistical analysis

Sample sizes (*n*, number of cells) for fluorescence microscopy were chosen based on our ability to detect decreases in bud neck localization in our previous studies (*Johnson et al., 2015*; *Weems and McMurray, 2017*). Similarly, sample sizes (*n*, number of particles) for single-particle EM analysis were chosen based on our ability to detect changes in septin complex subunit composition in our previous study of the Cdc3(G261V) mutation (*McMurray et al., 2011*), and sample sizes (*n*, number of replicate wells) for growth curve analysis were chosen based on our ability to detect changes in the growth rates of septin-mutant strains in our previous study (*Schaefer et al., 2016*). Each individual experiment was performed at least once, with all experimental samples and relevant controls prepared and analyzed in parallel. Analysis of effect sizes and generation of effect size plots were performed using DABEST (data analysis with bootstrap-coupled estimation) (*Ho et al., 2019*) via the online server at http://www.estimationstats.com. Other violin plots were generated with Prism 8.0 (Graphpad) using the 'medium' smoothing/kernel density setting.

## Acknowledgements

We thank Michelle Momany (University of Georgia, USA) for fruitful discussions about septin evolution and sharing unpublished reagents. The UCSF Chimera software package is developed by the Resource for Biocomputing, Visualization, and Informatics at the University of California, San Francisco (supported by NIGMS P41-GM103311). We also acknowledge the PICT-IBiSA (Institut Curie, Paris).

# Additional information

## Funding

| Funder | Grant reference number | Author |
|---|---|---|
| National Institute of General Medical Sciences | R00GM086603 | Michael A McMurray |
| National Institute of General Medical Sciences | R01GM124024 | Michael A McMurray |
| Alzheimer's Association | NIRGD-12-241119 | Michael A McMurray |
| Rare Genomics Institute | BeHEARD Initiative | Michael A McMurray |
| Agence Nationale de la Recherche | ANR-10-INSB-04 | Aurélie Bertin |
| Agence Nationale de la Recherche | ANR-10-LBX-0038 | Aurélie Bertin |
| National Science Foundation | MCB-1615138 | Anum Khan<br>Amy Galdfelter |

The funders had no role in study design, data collection and interpretation, or the decision to submit the work for publication.

## Author contributions

Courtney R Johnson, Marc G Steingesser, Investigation, Writing - review and editing; Andrew D Weems, Conceptualization, Formal analysis, Investigation, Writing - review and editing; Anum Khan, Investigation, Visualization, Writing - review and editing; Amy Gladfelter, Conceptualization, Supervision, Funding acquisition, Investigation, Visualization, Methodology, Writing - review and editing; Aurélie Bertin, Conceptualization, Formal analysis, Investigation, Visualization, Methodology, Writing - review and editing; Michael A McMurray, Conceptualization, Formal analysis, Supervision, Funding acquisition, Investigation, Visualization, Methodology, Writing - original draft, Project administration, Writing - review and editing

## Author ORCIDs

Michael A McMurray (iD) https://orcid.org/0000-0002-4615-4334

## Decision letter and Author response

Decision letter https://doi.org/10.7554/eLife.54355.sa1
Author response https://doi.org/10.7554/eLife.54355.sa2

# Additional files

## Supplementary files

- Supplementary file 1. Key resources table.
- Transparent reporting form

## Data availability

All data generated or analysed during this study are included in the manuscript and supporting files.

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
