## [Decision Letter]

**Acceptance summary:**

This manuscript combines examination of purified proteins and living cells with in silico modelling to investigate the roles of particular amino acid residues in the assembly of septin complexes on the background of fungal evolution. The presented insights will be of interest to a general biological audience because they advance our understanding of the regulation of septin filament assembly and the role of septin GTPase activity in this process.

**Decision letter after peer review:**

Thank you for submitting your article "Guanidine hydrochloride reactivates an ancient septin hetero-oligomer assembly pathway in budding yeast" for consideration by *eLife*. Your article has been reviewed by three peer reviewers, and the evaluation has been overseen by a Reviewing Editor and Anna Akhmanova as the Senior Editor. The following individuals involved in review of your submission have agreed to reveal their identity: Marian Farkasovsky (Reviewer #2).

Summary:

This is a neat example of how a fortuitous discovery – rescue of a septin Cdc10 mutant with GdnHCl – can lead to fundamental insights into the role of the GTPase activity in septin assembly. Aside of the evolutionary implications of the work, the findings of this manuscript provide a major mechanistic explanation on how GTPase activity drives the selection and exclusion of specific septin paralogs from higher assembly. This is particularly exciting as it provides a conceptual framework for understanding how a subset of septin paralogs may localize and function as non-canonical "free" homomers independently of their cognate partners, as it has been reported for Cdc10 in some fungi and SEPT9 in mammalian cells.

Essential revisions:

1) Subsection “Unbiased in silico modeling and phylogenetic analysis point to Thr302 as the site of Gdm binding in Cdc3”: All species with Arg mutation seems to belong to *Saccharomyces* clade, adding fungal phylogenetic tree might be valuable to the manuscript for better illustration of this point (maybe as a figure supplement).

2) There is some published evidence that subdenaturing concentrations of GdnHCl can stabilize protein (e.g. DOI: 10.1529/biophysj.104.044701). It will be useful to discuss this point in a few sentences. Did the authors test higher concentration of GdnHCl (0.1-0.4M)? Maybe higher concentration can "reactivate" the Cdc10-less septin complex expressed in *E. coli* and isolated in the absence of GdnHCl.

3) Another point which should be mentioned is the concentration of the protein. Formation of the Cdc3-Cdc11-Cdc12 filaments in vitro has been described (Farkasovsky et al., 2005). In that study, the authors used higher concentration of protein (1mg/ml) and protein dialysis in contrast to protein dilution. Assembly of Cdc3-Cdc11-Cdc12 filaments might be the question of concentration of properly folded complex. The authors should briefly mention this in the Discussion.

4) Did the authors see different elution profile in the presence of GdnHCl?

5) The implications of the findings for mammalian septins in light of the new order of assembly are quite exciting. The authors devote roughly a page, but it's worth expounding a bit more on SEPT9 occupying the central position in the hetero-octamer. They reference data of a SEPT7 mutation, which enhances the GTPase rate with concomitant shift from hetero-octamers to smaller complexes. It would be interesting to know what the authors think about the GTPases activity of SEPT9, which has been reported to be significantly faster than other septins, and the fact that SEPT9 and Cdc10 fall into the same ancient evolutionary clade. SEPT9 homo-dimerizes though the NC interface and hetero-dimerizes with SEPT7 through the G interface. How will the GTPase activities of SEPT9 and SEPT7 influence the apposing G interfaces and how would it be possible for SEPT9 to crash out of the hetero-octamer as potentially a free homomer? It would be interesting and useful to the septin community if the authors can comment with any insights into this possibility.

6) It would be helpful to describe the EM results in Figure 3 more quantitatively or at least give the readers a clear sense how efficiently the recombinant Cdc3, Cdc12, Cdc11 formed trimers, tetramers, pentamers, and hexamers in the presence of GdnHCl.

---

## [Author Response]

Essential revisions:1) Subsection “Unbiased in silico modeling and phylogenetic analysis point to Thr302 as the site of Gdm binding in Cdc3”: All species with Arg mutation seems to belong to Saccharomyces clade, adding fungal phylogenetic tree might be valuable to the manuscript for better illustration of this point (maybe as a figure supplement).

An excellent suggestion. Indeed, the fungal species in which the Arg was lost all fall in the family Saccharomycetaceae. We created a phylogenetic tree based on Cdc3 protein sequence and also included in it the broader taxonomic differences, to highlight the reviewer’s point, and added this panel to Figure 4—figure supplement 1.

2) There is some published evidence that subdenaturing concentrations of GdnHCl can stabilize protein (e.g. DOI: 10.1529/biophysj.104.044701). It will be useful to discuss this point in a few sentences. Did the authors test higher concentration of GdnHCl (0.1-0.4M)? Maybe higher concentration can "reactivate" the Cdc10-less septin complex expressed in *E. coli* and isolated in the absence of GdnHCl.

This is a fascinating reference of which we were unaware. Most interestingly, the literature suggests that both urea and GdnHCl operate via the same mechanism at subdenaturing concentrations to non-specifically stabilize proteins. The failure of urea to rescue *cdc10* mutants in vivois consistent with a more specific mechanism for GdnHCl, such as the model we propose of binding near Thr302 in Cdc3. We added wording to this effect to the section in the Results section. With regard to testing higher concentrations of GdnHCl on Cdc10-less septin complexes in vitro, we are intrigued by the idea, but if such experiments did indeed show a “reactivation” effect, this might represent a distinct mechanism unrelated to that responsible for *cdc10* rescue in vivo.

3) Another point which should be mentioned is the concentration of the protein. Formation of the Cdc3-Cdc11-Cdc12 filaments in vitro has been described (Farkasovsky et al., 2005). In that study, the authors used higher concentration of protein (1mg/ml) and protein dialysis in contrast to protein dilution. Assembly of Cdc3-Cdc11-Cdc12 filaments might be the question of concentration of properly folded complex. The authors should briefly mention this in the Discussion.

This is an important point that fits well with other related concepts. We added a paragraph to the Discussion that explains how various factors can contribute to filament formation in vitro by increasing the concentration of properly folded complex and includes the point mentioned by the reviewer.

4) Did the authors see different elution profile in the presence of GdnHCl?

Yes. In the presence of GdnHCl a “new” SEC elution peak appeared between the aggregates found in the void volume and the trimers and dimers found in the absence of GdnHCl. It was this peak that formed filaments upon dilution to low salt, which fits with the idea that this fraction contained the hexamers that were competent to form filaments. We have now added these data to Figure 3 and mention of these findings to the body text, and we thank the reviewer for the suggestion.

5) The implications of the findings for mammalian septins in light of the new order of assembly are quite exciting. The authors devote roughly a page, but it's worth expounding a bit more on SEPT9 occupying the central position in the hetero-octamer. They reference data of a SEPT7 mutation, which enhances the GTPase rate with concomitant shift from hetero-octamers to smaller complexes. It would be interesting to know what the authors think about the GTPases activity of SEPT9, which has been reported to be significantly faster than other septins, and the fact that SEPT9 and Cdc10 fall into the same ancient evolutionary clade. SEPT9 homo-dimerizes though the NC interface and hetero-dimerizes with SEPT7 through the G interface. How will the GTPase activities of SEPT9 and SEPT7 influence the apposing G interfaces and how would it be possible for SEPT9 to crash out of the hetero-octamer as potentially a free homomer? It would be interesting and useful to the septin community if the authors can comment with any insights into this possibility.

We are happy to expound upon this line of reasoning. In the models we favor (which are admittedly biased by our prior work with Cdc10), SEPT9 cannot “crash out” of a hetero-octamer, because it hydrolyzes GTP prior to incorporation into a hetero-oligomer, and is catalytically inactive thereafter. The choice by SEPT9 of septin hetero-oligomerization or not would, according to our models, depend more upon the nucleotide state of SEPT7 during de novo assembly, which is supported by existing data from the literature. We added a paragraph to this effect in the Discussion.

6) It would be helpful to describe the EM results in Figure 3 more quantitatively or at least give the readers a clear sense how efficiently the recombinant Cdc3, Cdc12, Cdc11 formed trimers, tetramers, pentamers, and hexamers in the presence of GdnHCl.

We have added a histogram/frequency distribution of particle lengths that illustrates the efficiency of assembly/stability of complexes formed in the presence and absence of GdnHCl.